# An achromatic metafiber for focusing and imaging across the entire telecommunication range

Haoran Ren [1,2,10✉], Jaehyuck Jang[3,10], Chenhao Li[2,10], Andreas Aigner[2], Malte Plidschun [4,5], Jisoo Kim[4,5], Junsuk Rho [3,6,7✉], Markus A. Schmidt[4,5,8✉] & Stefan A. Maier [1,2,9✉]

Dispersion engineering is essential to the performance of most modern optical systems including fiber-optic devices. Even though the chromatic dispersion of a meter-scale single-mode fiber used for endoscopic applications is negligible, optical lenses located on the fiber end face for optical focusing and imaging suffer from strong chromatic aberration. Here we present the design and nanoprinting of a 3D achromatic diffractive metalens on the end face of a single-mode fiber, capable of performing achromatic and polarization-insensitive focusing across the entire near-infrared telecommunication wavelength band ranging from 1.25 to 1.65 μm. This represents the whole single-mode domain of commercially used fibers. The unlocked height degree of freedom in a 3D nanopillar meta-atom largely increases the upper bound of the time-bandwidth product of an achromatic metalens up to 21.34, leading to a wide group delay modulation range spanning from −8 to 14 fs. Furthermore, we demonstrate the use of our compact and flexible achromatic metafiber for fiber-optic confocal imaging, capable of creating in-focus sharp images under broadband light illumination. These results may unleash the full potential of fiber meta-optics for widespread applications including hyperspectral endoscopic imaging, femtosecond laser-assisted treatment, deep tissue imaging, wavelength-multiplexing fiber-optic communications, fiber sensing, and fiber lasers.

[1] School of Physics and Astronomy, Faculty of Science, Monash University, Melbourne, Victoria 3800, Australia. [2] Chair in Hybrid Nanosystems, Nanoinstitute Munich, Faculty of Physics, Ludwig Maximilian University of Munich, Munich 80539, Germany. [3] Department of Chemical Engineering, Pohang University of Science and Technology (POSTECH), Pohang 37673, Republic of Korea. [4] Leibniz Institute of Photonic Technology, 07745 Jena, Germany. [5] Abbe Center of Photonics and Faculty of Physics, FSU Jena, 07745 Jena, Germany. [6] Department of Mechanical Engineering, Pohang University of Science and Technology (POSTECH), Pohang 37673, Republic of Korea. [7] POSCO-POSTECH-RIST Convergence Research Center for Flat Optics and Metaphotonics, Pohang 37673, Republic of Korea. [8] Otto Schott Institute of Material Research, FSU Jena, 07745 Jena, Germany. [9] Department of Physics, Imperial College London, London SW7 2AZ, UK. [10] These authors contributed equally: Haoran Ren, Jaehyuck Jang, Chenhao Li. ✉email: Haoran.Ren@monash.edu; jsrho@postech.ac.kr; Markus.Schmidt@leibniz-ipht.de; Stefan.Maier@monash.edu

Optical fibers are of great technological importance due to their unique features such as flexible handling, strong light confinement, and efficient light transportation over large distances, leading to a multitude of applications in modern optics, including fiber communications[1], optical trapping[2,3], nonlinear light generation[4], sensing[5], and endoscopic imaging[6–8]. In particular, flexible fiber endoscopes allow for the scanning imaging of internal tissues in vivo environments for medical diagnosis. During fiber imaging, scattered and emitted light from the object are collected by a bulky gradient-refractive-index lens[9–11], a refractive ball lens[12], or recently 3D-printed free-form microlenses[13,14] being implemented onto the end face of a fiber. However, constraints in the geometric shape and index profile of these lenses result in strong group delay that cannot be compensated, leading to significant chromatic aberration that obscures optical images over a broad wavelength range of interest. To date, control over the chromatic dispersion of a focusing or imaging lens located on the end face of an optical fiber is not possible, and the fiber output is handled by, for instance, a bulky doublet lens (made of glasses with different material dispersions) in free-space[15], which inevitably limits the scope of fiber-based applications.

Meta-optics, which allows the arbitrary shaping of light by using ultrathin metasurfaces, offers an unprecedented platform to realize optical lenses[16], vortex plates[17,18], holograms[19–21], and vector waveplates[22–24] with outstanding performance. A metalens, an ultrathin metasurface lens, not only allows for diffraction-limited light focusing, but also for a correction of possible aberrations without using additional optical elements[25–33]. Achromatic diffractive metalenses fabricated on planar substrates have recently been demonstrated at fiber-relevant telecommunication bands[33–35], exhibiting a significantly reduced form factor in comparison to a conventional achromatic doublet. However, for acquiring various group delay responses previous achromatic metalenses inevitably use a complex design of meta-atoms, i.e., the unit cells of a metasurface, with a variety of geometries[25–35]. More critically, the achromatic performance of previous metalenses is rather limited by the modulation capability of the group delay in their complex meta-atom designs. Thus far, the maximal upper bound of the time-bandwidth product (TBP)[36]—the product of achievable time delay and spectral bandwidth that sets the fundamental bandwidth limit of an achromatic metalens—has been limited to 11.5 (Fig. S1).

On the other hand, for advanced wavefront manipulation of the fiber output, interfacing functional metasurfaces with optical fibers are strongly demanded. Even though the implementation of metasurfaces on fiber end faces that enable light focusing and bending has been realized via ion-beam lithography[37,38] and hydrofluoric chemical-etching techniques[39], these fabrication methods fail to create arbitrary structures for efficient light modulation, thus limiting their photonic functionality and applications. Alternatively, electron-beam lithography[40] and nanoimprinting[41] techniques provide fabrication resolution in the deep subwavelength regime, nevertheless, facing compelling challenges to prepare the fiber end face for planar surface patterning and predesigned-pattern transfer. 3D laser nanoprinting, based on two-photon polymerization, offers an ideal platform for optically implementing 3D diffractive microstructures on a fiber facet[13,14,42–45], paving the way to functionalize optical fibers for different photonic applications.

## Results

Here we demonstrate the design and 3D nanoprinting of an achromatic metafiber through interfacing a 3D achromatic diffractive metalens with a commonly used telecommunication single-mode fiber (SMF-28). Owing to the degraded polarization purity subject to environmental perturbations in a SMF-28, our achromatic metalens was designed to be polarization insensitive. The unlocked height degree of a 3D achromatic metalens largely increases the upper bound of the TBP to 21.34 (Fig. S1), thus expanding the modulation range of the group delay from −8 to 14 fs. Meanwhile, subwavelength nanopillar meta-atoms of the achromatic metalens exhibit strong birefringence, capable of imprinting a hyperbolic lens profile via the geometric phase. As a result, our designed and constructed 3D achromatic metalens on a SMF, named achromatic metafiber (Fig. 1), enables achromatic and diffraction-limited focusing over the entire near-infrared telecommunication band ranging from 1.25 to 1.65 μm, which particularly represents the whole single-mode operation regime of this type of fiber. In addition, we employed the achromatic metafiber for fiber-optic confocal scanning imaging without involving a standard microscope, demonstrating that our approach offers an unprecedented solution to establish an ultra-compact and ultrabroadband confocal endoscopic system.

**Time-bandwidth product of a 3D achromatic metalens**. Unlike 2D planar metalenses that exhibit a limited modulation range of group delay, our 3D metalens offers a degree of freedom in height that can provide a large tuning range of group delay. As a 3D nanopillar meta-atom in principle represents a truncated waveguide, its group delay response linearly increases with the height of the nanopillar (Supplementary Note 1 and Fig. S2). The slope of the group delay with respect to its height can be altered by the transverse dimensions of the nanopillar, leading to different effective refractive indices ($n_{\text{eff}}$) of the waveguide modes. This largely expanded group delay range holds great potential to provide better performance of an achromatic metalens, which can be understood by the TBP of a 3D achromatic metalens[36]. The upper bound of the TBP imposes a fundamental bandwidth limit on an achromatic metalens[36,46], which is generally written as $\kappa \geq \Delta T \Delta \omega$, where $\kappa$ is a dimensionless quantity, and $\Delta T$ and $\Delta \omega$ are the time delay and spectral bandwidth of the achromatic metalens, respectively (Fig. S3).

For dielectric planar metalenses[26–33] that are based on waveguide-type meta-atoms with a fixed height $H$, the value of the upper bound TBP is given by (Supplementary Note 2):

$$\kappa = \frac{\omega_c}{c} H \cdot (n_{\text{eff}}^{\max} - n_{\text{eff}}^{\min}) \tag{1}$$

where $n_{\text{eff}}^{\max}$ and $n_{\text{eff}}^{\min}$ are the maximum and minimum effective refractive indices of the metalens at a central frequency, and $\omega_c$ and $c$ are the angular frequency at a central frequency and the speed of light, respectively. Equation 1 can be regarded as the phase difference between the waveguide modes with the slowest and fastest group velocities imposed by nanopillar meta-atoms. Both the propagation constant, which is the product of the effective refractive index and the vacuum wavenumber, and the height of a meta-atom determine the phase difference. For our 3D nanopillar meta-atoms, the maximal phase delay is achieved when both the effective refractive index (controlled by the transverse size of a nanopillar waveguide) and the height of a meta-atom are maximal (Fig. S4A). On the other hand, the minimal phase delay is obtained when both quantities are minimal (Fig. S4B). Hence, the upper bound of TBP κ of a 3D metalens can be formulated as:

$$\kappa = \frac{\omega_c}{c} \left( H_{\max} n_{\text{eff}}^{\max} \right) - \frac{\omega_c}{c} \left( H_{\min} n_{\text{eff}}^{\min} + (H_{\max} - H_{\min}) n_b \right)$$
$$= \frac{\omega_c}{c} (H_{\max} \cdot \Delta n_{\text{eff}}^{\max} - H_{\min} \cdot \Delta n_{\text{eff}}^{\min}) \tag{2}$$

where $H_{\max}$ and $H_{\min}$ are the maximum and minimum height of 3D meta-atoms, respectively; $\Delta n_{\text{eff}}^{\max} = n_{\text{eff}}^{\max} - n_b$ and

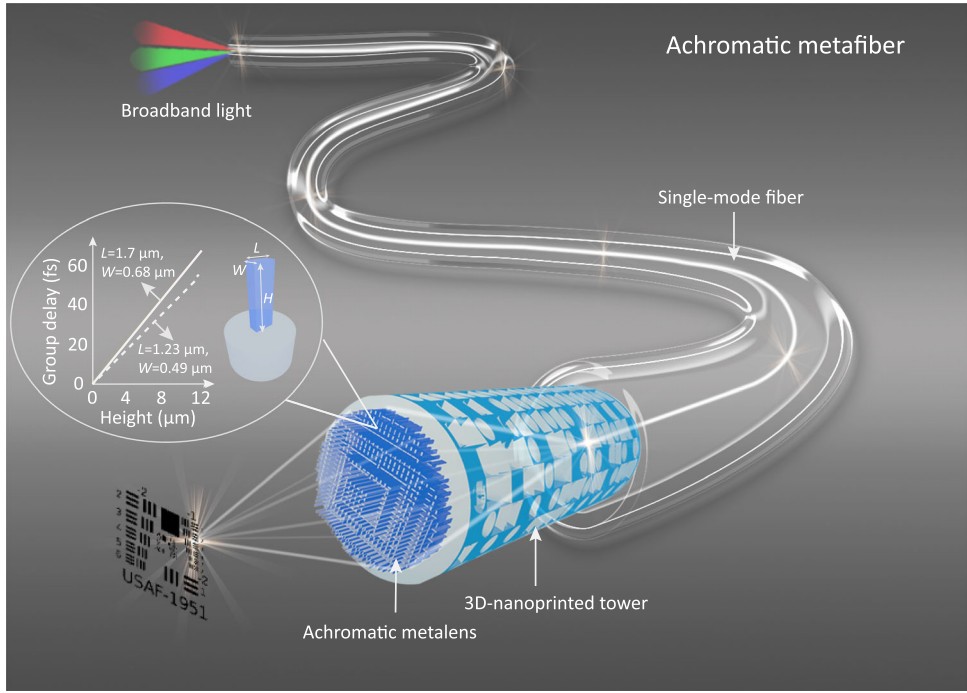

**Fig. 1 Principle of an achromatic metafiber used for achromatic focusing and imaging.** An achromatic metalens located on top of a 3D-printed hollow tower (used for fiber-beam expansion) was interfaced with a single-mode fiber via 3D laser nanoprinting. Inset: an enlarged 3D nanopillar meta-atom (height: *H*, length: *L*, width: *W*), the height of which offers a large modulation range of group delay.

$\Delta n_{\text{eff}}^{\min} = n_{\text{eff}}^{\min} - n_b$, where $n_b$ is the refractive index of the surrounding background hosting meta-atoms. It is obvious to see that the unleashed degree of freedom in height could largely extend the upper bound of the TBP as compared to previous results in earlier papers (Supplementary Table 1). More critically, a broad range of group delays can be easily accessed by the height of 3D meta-atoms, without the need of significantly changing the geometry of meta-atoms as in earlier papers (Fig. S1).

**Design of an achromatic metafiber.** To achieve efficient wave-front manipulation for the output beam of a SMF, we designed and printed a hollow tower structure on the end face of the SMF in order to expand its output in free-space, as well as to hold the achromatic metalens on top. As such, the SMF output of broadband light gets expanded significantly (with the mode field diameter increasing from around 9.5 to 100 μm) through free-space propagation in the hollow tower structure (Fig. 1). For the design of an achromatic metafiber, the required phase and group delay of an achromatic metalens were theoretically calculated based on a spherical lens profile at a nominal wavelength of 1650 nm (Supplementary Note 3). Without loss of generality, an achromatic metalens was designed to have a lens diameter of 100 μm, a numerical aperture (NA) of 0.2, and a spectral bandwidth of 400 nm ranging from 1250 to 1650 nm (Fig. S5A). The fiber wavefront was corrected by assuming the SMF output to have a Gaussian beam profile as well as a divergent phase profile with a negative focus (Supplementary Note 4). This divergent fiber wavefront leads to a reduced NA of 0.12 of the achromatic metafiber. The maximal bandwidth of our designed metalens was determined from our constructed 3D meta-atom design library based on the TBP consideration (Supplementary Note 5). The basic properties of an achromatic metalens including lens diameter, bandwidth, NA, and focal length are physically correlated and theoretically bounded by the TBP limit (Figs. S5B and 5C). It should be mentioned that, owing to this expanded TBP limit by

our 3D meta-atoms, we can have more flexible choices on the NA and bandwidth of an achromatic metalens. Unlike a metalens optimized for a single wavelength, the NA of which can be increased by either enlarging the lens diameter or reducing the focal length, here we have to consider the fundamental trade-off between the bandwidth and NA of an achromatic metalens based on the TBP consideration (Fig. S5B).

A design library that contains phase and group delay in the cross-polarization response of 3D polymer nanopillars was constructed based on our in-house rigorous coupled-wave analysis model (see the "Methods" section for details). According to the principle of the Pancharatnam-Berry phase[47], a geometric phase is imparted to cross-polarized light that is scattered from a birefringent nanopillar by twice the amount of the in-plane rotation angle ($\alpha$). In our simulation, we considered 3D nanopillars made of IP-L 780 polymer (Nanoscribe GmbH, Germany) with a constant period of $P = 2.2$ μm, length of $L = 0.85$–1.7 μm, height of $H = 8.5$–13.5 μm, and an in-plane aspect ratio of $R = 0.3$–0.6 (Fig. 2A–C). Notably, the subwavelength length and width of 3D nanopillars give rise to a strong birefringent response, which can be used to imprint a hyperbolic lens profile at a fixed wavelength. Meanwhile, the height of nanopillars in a range much larger than the wavelength results in significantly expanded group delay and TBP responses, as predicted by Eq. 2. The resultant library data was plotted in phase and group delay space (blue and red dots in Fig. 2D). Here each dot corresponds to the simulation results of a specific 3D nanopillar with a distinctive combination of geometric parameters, including height and transverse dimensions. It should be mentioned that the π-phase difference between the nanopillars rotated at 0 and 90 degrees enables us to fill most of the phase and modified group delay space (gray dots in Fig. 2D, each dot refers to a phase and group delay pair that is required in an achromatic lens design), in contrast being difficult to achieve for meta-atoms that are based on either resonant or propagation phase responses[25].

To explicitly show that our 3D meta-atom library can fulfill the design of an achromatic metalens with NA = 0.2, we compared

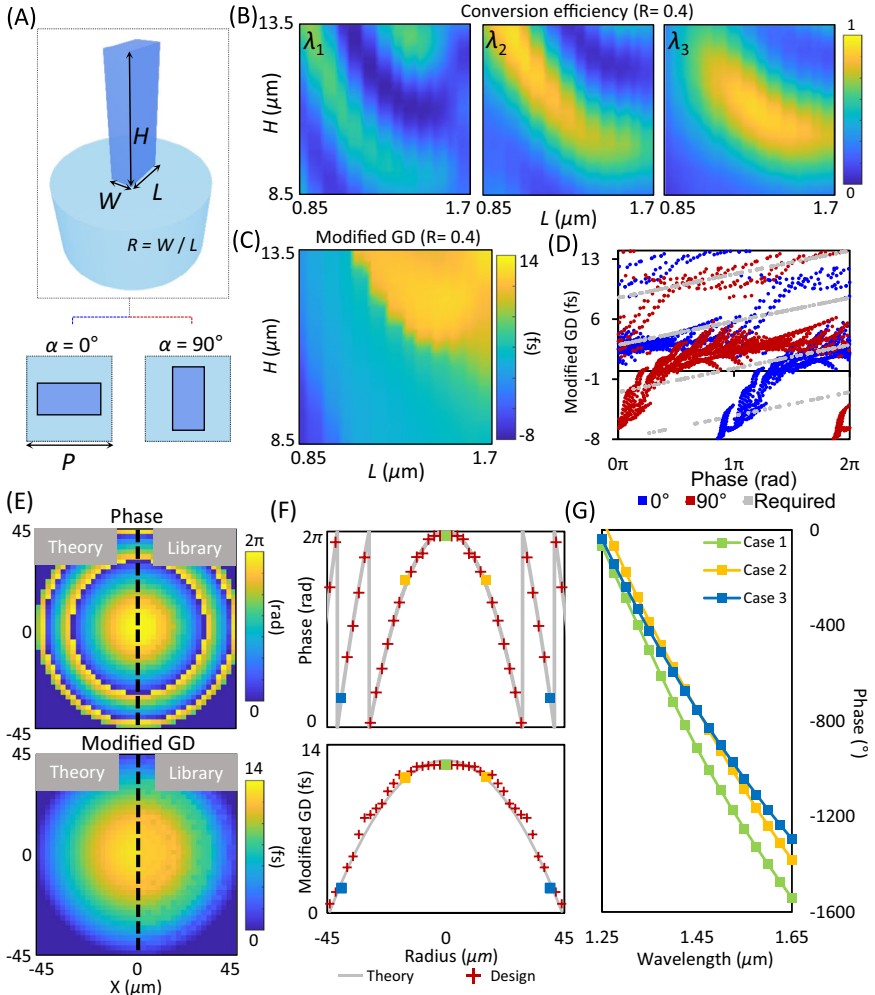

**Fig. 2 A 3D meta-atom library for the design of an achromatic metalens. A** Schematic of a low-index IP-L nanopillar-based 3D meta-atom. P: pitch distance; H: height; L: length; W: width; R: in-plane aspect ratio. **B** Polarization conversion efficiency of nanopillar waveguides with varied parameters of height and length at three different wavelengths of $\lambda_1 = 1250$ nm, $\lambda_2 = 1450$ nm, and $\lambda_3 = 1650$ nm, respectively. The in-plane aspect ratio was fixed to $R = 0.4$. **C** Group delay response for $R = 0.4$. **D** The designed phase and group delay library. Gray dots: required pairs by an achromatic lens design in Fig. S5A; blue dots: simulated pairs based on the nanopillars with an orientation of 0 degree; red dots: simulated pairs based on the nanopillars with an orientation of 90 degrees. **E**, **F** Theoretical and numerical design of the phase and group delay responses of an achromatic metalens, based on the diffraction theory (left half) and selected 3D meta-atoms (right half), respectively. **F** presents the phase and group delay profiles along the radial direction of the achromatic metalens, respectively, with gray lines for the theoretical design and cross dots for the numerical results based on selected 3D meta-atoms. The colored squares label three different nanopillar meta-atoms (Case 1: $L = 1.625$ μm; $R = 0.5$; $H = 13.5$ μm; $\alpha = 0°$; Case 2: $L = 1.5$ μm; $R = 0.5$; $H = 13.25$ μm; $\alpha = 0°$; Case 3: $L = 1.25$ μm; $R = 0.4$; $H = 11$ μm; $\alpha = 90°$) that have different group delay responses. **G** Phase response of cross-polarized light for the three different meta-atoms labeled in **F**.

theoretically required and the library-provided phase and modified group delay profiles in Fig. 2E, both of which exhibit a good consistency. Figure 2F plots the phase and modified group delay profiles along the radial direction of the achromatic lens (Supplementary Table 2). Instead of using 2D meta-atoms with a variety of geometries to realize different group delays[24–33], we use both the height and transverse size of 3D nanopillars to acquire a large dynamic range of group delay responses. As an example, we show that three nanopillars of different heights exhibit large differences in their group delay responses (2.1 fs, 11.3 fs, and 12.3 fs), which can be understood from their capabilities of phase compensation across the wavelengths of interest (Fig. 2G), the entire group delay data of which is given in Fig. S6. Therefore, we selected nanopillars that are closest matched with the phase and group delay requirements to design an achromatic metafiber. Due to the fact that the SMF output features random polarization perturbations that are influenced by environmental changes such

as fiber bending, it is highly demanded to develop an achromatic metalens that is insensitive to the polarization of incident light. We achieved the polarization insensitivity in our achromatic metalens by setting the in-plane rotation angles of the 3D nanopillars to be either 0 or 90 degrees, thus removing the circular polarization dependence of a geometric metasurface[48] (Supplementary Note 6).

**Characterization of a 3D achromatic metalens on planar substrate**. For an experimental demonstration of achromatic focusing of light, the designed 3D achromatic metalens (clear aperture size of 100 μm) was first printed onto a silica substrate using 3D laser nanoprinting technology (Fig. 3A, see the "Methods" section for details). High aspect ratio nanopillars (the maximal ratio between the height and transverse dimension reaches up to 33.75) with varied heights ranging from 8.5 to 13.5 μm have been

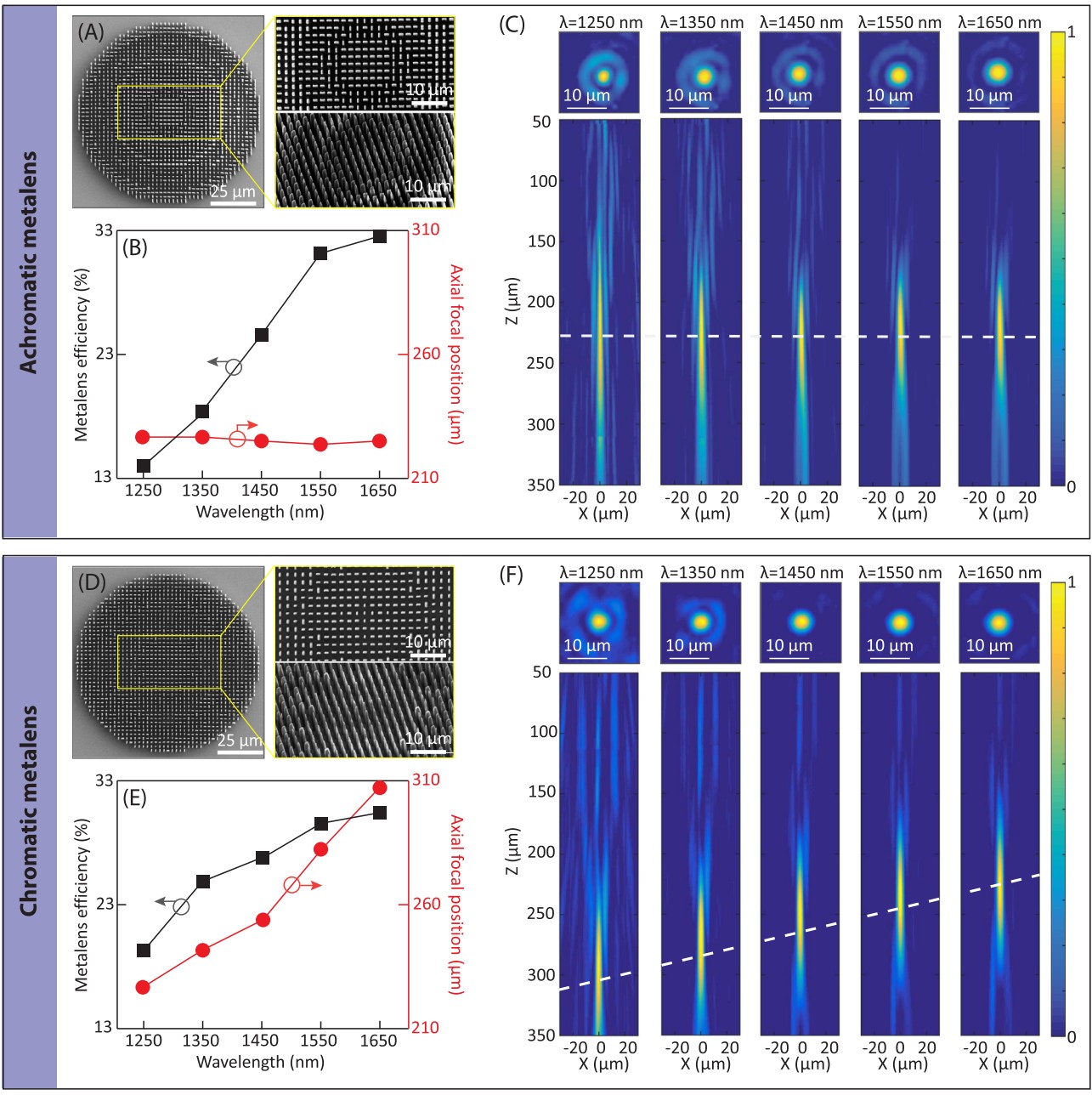

**Fig. 3 Experimental characterization and comparison of an achromatic metalens and a chromatic metalens on a glass substrate. A** SEM images of the achromatic metalens, with magnified area images in the top (top inset) and the oblique (bottom inset) views. **B** Experimentally characterized metalens efficiency and axial focal positions of the achromatic metalens on planar substrate at different wavelengths. **C** Experimentally characterized point-spread functions of the achromatic metalens in both transverse (top) and longitudinal (bottom) planes at five different wavelengths of 1250 nm, 1350 nm, 1450 nm, 1550 nm, and 1650 nm, respectively. **D**–**F** The counterparts of (**A**–**C**) for the case of a chromatic metalens on planar substrate, which has the same spherical lens profile as the achromatic metalens but without the group delay compensation.

successfully printed, as shown in the side-view scanning electron microscope (SEM) image (Fig. 3A). To characterize the achromatic metalens, a tunable laser beam that is based on a supercontinuum laser source (SUPERK FIANIUM, NKT Photonics) and an infrared wavelength selector (SUPERK SELECT, NKT Photonics) was incident on the fabricated achromatic metalens. The metalens on planar substrate was illuminated by a collimated beam without a divergent wavefront (Fig. S7). We experimentally characterized the efficiency of our 3D-nanoprinted achromatic metalens (transmission efficiency multiplied by the focusing efficiency, the data is presented in Supplementary Table 3), giving rise to efficiency values of

14.13%, 18.11%, 24.58%, 30.87%, and 32.01% for incident wavelengths of 1250, 1350, 1450, 1550, and 1650 nm, respectively (Fig. 3C). We have to mention that our meta-atoms were numerically optimized to exhibit strong polarization conversion efficiency particularly at the nominal wavelength of 1650 nm for the metalens design, exhibiting higher conversion efficiency than other wavelengths (Fig. 2B). As such, the metalens efficiency appears to be frequency-dependent and decreases by increasing frequency (Fig. 3B). In addition, the measured axial focal positions of the achromatic metalens remain close to identical for different wavelengths (Fig. 3C and D), verifying that our printed metalens is achromatic in nature.

As a result, the fabricated achromatic metalens on the planar substrate was characterized having an NA of 0.2, leading to an achromatic and polarization-insensitive focus over the entire telecommunication wavelength range (Fig. S8). The size of the experimentally characterized achromatic focal spot is consistent with a diffraction-limited focus, showing a full width at half maximum (FWHM) of 3.03 μm (the diffraction-limited case of 3.05 μm), 3.83 (3.52), 3.79 (3.82), 3.84 (4.01), and 4.24 (4.32) for an incident wavelength of 1.25, 1.35, 1.45, 1.55, and 1.65 μm, respectively (Supplementary Table 4). We evaluated how an additional non-uniform amplitude distribution in the cross-polarization component affects the diffraction-limited focusing of our metalens. We simulated and compared two cases of a spherical lens function (used by our metalens) with and without a non-uniform amplitude distribution extracted from our designed metalens in Fig. 2E, the results are shown in Fig. S9. It indicates that the non-uniform amplitude does not degrade the diffraction-limited focus, exhibiting a same FWHM, a similar focal length and a slightly higher focusing efficiency (due to the complex-amplitude modulation) as compared to the case of a uniform amplitude profile. As such, the negligible deviation of the measured FWHM values with respect to the diffraction-limited ones might result from other factors including an insufficient sampling of a lens profile (Supplementary Note 7). On the other hand, for comparison, we carried out a control experiment by printing a chromatic metalens with the same size and lens profile, but without the group delay compensation (Fig. 3D, and Fig. S10). It is obvious to see that the chromatic metalens also leads to a diffraction-limited focus with slightly higher efficiency (Fig. 3E, Supplementary Table 5), however, suffering from strong chromatic dispersion with the focal position axially shifting as a function of the incident wavelength (Fig. S11 and Fig. 3E and F). More specifically, the focal position shifts from 225 to 306 μm when the incident wavelength changes from 1.65 to 1.25 μm. It should be mentioned that for both achromatic and chromatic metalenses, a small shift of focal position ($f = 225$ μm) with respect to the nominal design ($f = 250$ μm) at a wavelength of 1.65 μm was observed, which is due to the spatial sampling of a lens phase profile (Fig. S12 and Supplementary Note 7).

**Fabrication and characterization of an achromatic metafiber.** To demonstrate an achromatic metafiber based on this principle, we experimentally interfaced an achromatic metalens with an SMF and characterized the metafiber focusing performance. For this purpose, a 2-m long SMF (SMF-28, Thorlabs) with a small core (mode field diameter 9.5 μm at 1450 nm) was first cleaved in order to obtain a flat end face. To increase the mode field diameter of the SMF output for an efficient wavefront manipulation on-fiber, a hollow tower structure of 525 μm height, 120 μm diameter, and 10 μm side-wall thickness was printed on the cleaved SMF (Fig. 4A and B). The height of the hollow tower structure was determined from the expanded fiber output having a mode field diameter large enough to cover the whole metalens. To avoid the fiber beam hitting the side-wall of the tower, the hollow tower was designed to have a diameter slightly larger than the mode field diameter of the fiber output at the position of the metalens. A close-up SEM image verifies the successful fabrication of the designed 3D achromatic metalens on the fiber tower (Fig. 4C). During the 3D nanoprinting, the SMF end face was centered precisely with respect to the laser beam (see the "Methods" section for more details on the fabrication process). The focusing performance of the 3D-nanoprinted achromatic metafiber was characterized by coupling light from a supercontinuum laser source into the fiber sample and placing the metafiber tip on a 3D piezo nanopositioning stage (P-563,

Physikinstrumente). We measured the transverse beam profile at different axial positions (axial step size of 0.5 μm with a 300 μm travel range) with an imaging objective (Olympus 100x, NA = 0.8). The experimentally obtained axial profiles of the metafiber foci at five different wavelengths of 1.25, 1.35, 1.45, 1.55, and 1.65 μm are presented (Fig. 4D), revealing the achromatic nature of the metafiber focus (the focal positions remain nearly constant for all the wavelengths within the interested bandwidth), exhibiting an effective NA of 0.12. The experimentally obtained FWHMs of the achromatic metafiber focus were 4.97 μm (the diffraction-limited case was 5.41 μm), 5.45 (5.83), 5.81 (6.23), 6.27 (6.63), and 6.74 (7.13) for incident wavelengths of 1.25, 1.35, 1.45, 1.55, and 1.65 μm, respectively (Supplementary Table 6). The slightly smaller FWHMs of our measured results compared to the diffraction-limited case possibly result from a small over-compensation of the divergent fiber wavefront.

**Achromatic metafiber imaging.** Achromatic imaging using a flexible fiber is the ultimate goal of the developed achromatic metafiber. Even though metalenses offer a new route to miniaturize fiber-optic imaging systems, severe chromatic aberrations prohibit them from being used for multiwavelength and broadband imaging[25]. In order to explicitly demonstrate the advanced performance of an achromatic metafiber for broadband imaging, we first simulated and compared imaging results of achromatic and chromatic metalens-implemented SMFs. In particular, the SMFs work as confocal microscope pinholes[49], where the implemented metalenses on-top miniaturize the whole confocal imaging system, thus providing greater flexibility and reducing alignment problems. Owing to the confocal imaging principle[50], the metalens collects object signals of different wavelengths at different focal planes through convoluting diffraction-limited point-spread-functions with the corresponding object signals at different planes (Fig. 5A–D). In this case, an achromatic metafiber achieves sharp in-focus imaging responses for broadband illumination, resulting from the constant focal positions of different wavelengths (Fig. 5A and B). In contrast, the chromatic metalens reduces the intensity and blurs the image responses of broadband illumination, since the axial and lateral chromatic aberrations affect both image collection efficiency and the image contrast in the scanned confocal images, respectively (Fig. 5C and D).

Here, we applied the 3D-nanoprinted achromatic metafiber for fiber-optic confocal scanning imaging without involving any other microscope components. The optical setup used for the measurement of the fiber-optic confocal imaging is schematically illustrated in Fig. S13. Transmitted light under a broad illumination from a moving object placed on a 3D piezo nanopositioning stage (P-563, Physikinstrumente) was directly collected by the achromatic metafiber (the metalens side). At its other end, the intensity of fiber transmitted light signals was measured by a Ge-amplified photodetector (PDA50B2, Thorlabs). In our experiment, we used a standard United States Air Force (USAF) 1951 resolution chart to investigate the resolution limit of our achromatic metafiber-based confocal imaging. Consequently, sharp images of different resolution targets (from 11.05 μm (Group 5 Element 4) to 8.77 μm (Group 5 Element 6)) were successfully obtained from the achromatic metafiber over the entire telecommunication wavelengths spanning from 1.25 to 1.65 μm (Fig. 5E). Furthermore, we compared the imaging performance of the achromatic metafiber with a chromatic metafiber through scanning the same target area, which results in blurred image responses (Fig. 5F), as well as significantly reduced image intensity and contrast at different wavelengths (except for the nominal design wavelength of 1.65 μm) (Fig. 5G). Our experiment clearly shows that the flexible achromatic metafiber

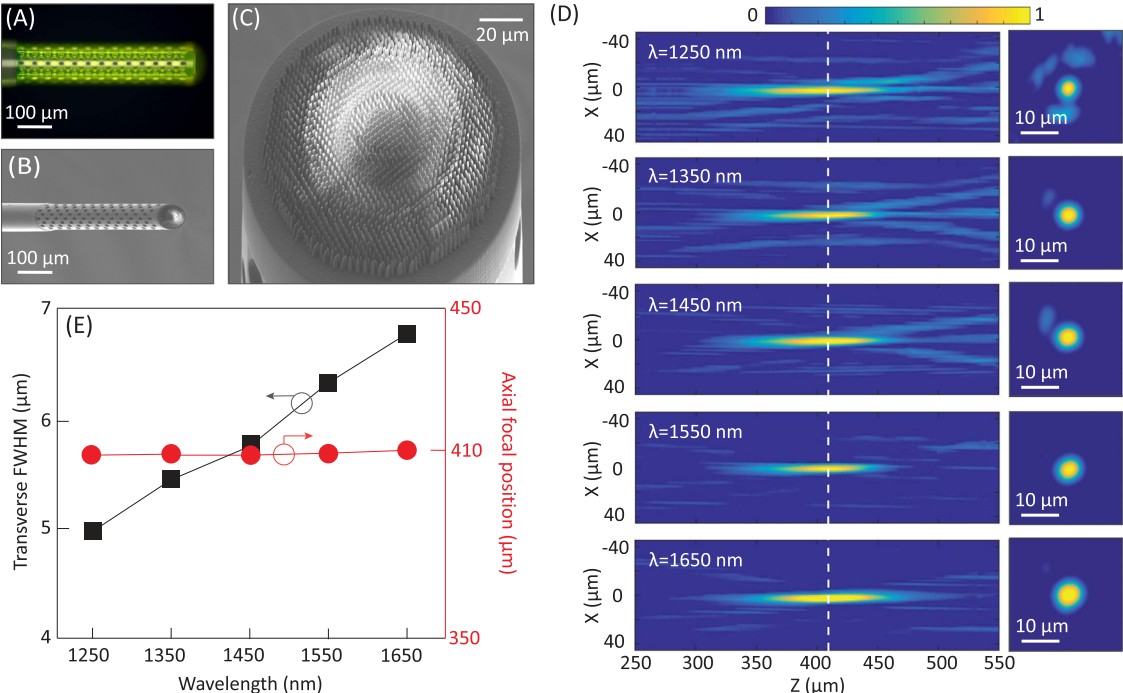

**Fig. 4 Focusing performance of a 3D-nanoprinted achromatic metafiber. A–C** Optical (**A**) and SEM (**B**, **C**) images of the achromatic metafiber.
**D** Experimentally characterized point-spread functions of the achromatic metafiber in both longitudinal (left) and transverse (right) planes at five different wavelengths of 1250, 1350, 1450, 1550, and 1650 nm, respectively. **E** Experimentally characterized transverse FWHM and the axial focal positions of the achromatic metafiber at different wavelengths.

generates in-focus images under broadband light illumination spanning the entire telecommunication regime, with a spatial resolution up to 4.92 μm (Fig. 5H). These results suggest that 3D meta-optics is capable of not only miniaturizing endoscopic imaging systems, but also to extending their general wavelength coverage by allowing high image quality under broadband illumination. In addition, although the telecommunication wavelength range was typically avoided due to the presence of a water overtone absorbance peak, it allows deep tissue imaging with large penetration depths[51]. This longer near-infrared wavelength range has facilitated higher imaging clarity, due to its exponentially reduced photon scattering and autofluorescence[51]. As such, we envision that our achromatic metafiber can be fabricated on a distal-scanning endoscope for practical endoscopic imaging[52], in which a fiber-based beamsplitter can be used to separate the excitation and fluorescence signals from a fluorescence sample (Fig. S14). Alternatively, if the excitation wavelength is beyond the SMF bandwidth, a microstructured fiber can be used[53], where the excitation and collection light signals are transported through individual guiding domains.

## Discussion

We have demonstrated the compensation of the chromatic dispersion of a fiber-integrated lens via 3D-nanoprinted meta-optics, opening up the possibility to create a large modulation range of the TBP and the group delay. Fixing the in-plane rotation angles of nanopillar meta-atoms to 0 and 90 degrees allows for the 3D achromatic metalens to be insensitive to the incident polarization of light. We have experimentally characterized the focusing performance of a miniaturized 3D-nanoprinted achromatic metafiber, which exhibits an achromatic focus with a constant focal length over the entire telecommunication and single-mode domain of a commercial SMF from 1.25 to 1.65 μm. It should be mentioned that previous focusing devices implemented on the

ends of fibers[54,55] possess a large depth of focus that allows their foci of different wavelengths overlap, but these devices are with much lower NAs (e.g., NA = 0.085[54] and NA = 0.0665[55]) and do not have the dispersion-compensation capabilities. It can be anticipated that a strong focal shift can be observed when the NAs of their devices are increased to the order of our work. In addition, to determine the pitch distance of meta-atoms, we have considered the trade-off between the near-field coupling and the lens sampling rate. We set a constant pitch of 2.2 μm in the metalens through balancing (i) high cross-polarization conversion efficiency; (ii) sufficient sampling of a lens function with high efficiency; (iii) negligible near-field coupling (Supplementary Note 7). As such, the efficiency loss of our achromatic metalens can be attributed to two major sources: (i) a non-negligible co-polarization component in some of selected meta-atoms used for covering a broad span of group delay; (ii) insufficient sampling of a lens profile that reduces the focusing efficiency (Supplementary Note 7).

Furthermore, we have shown direct confocal scanning imaging with our developed flexible achromatic metafiber, allowing to produce in-focus sharp images (spatial resolution up to 4.92 μm) under broadband light illumination. We envision that our compact and flexible achromatic metafiber could replace a conventional optical table-based confocal imaging system, thus allowing for greater flexibility and reduced alignment problems. Apart from high-performing on-axis imaging, our demonstrated achromatic metalens also exhibits a good performance for off-axis imaging up to ±7.5 degrees that correspond to a maximal field of view of 104 μm (@1650 nm) in the focal plane (Supplementary Note 8 and Fig. S15). We believe, however, there is a general technical challenge in the fiber-based wide-field imaging, as the off-axis light path could induce an angular offset with respect to the fiber axis (the nominal emission and collection direction of a SMF). This could reduce the collection efficiency of off-axis imaging and limit the field-of-view of the achromatic metalens

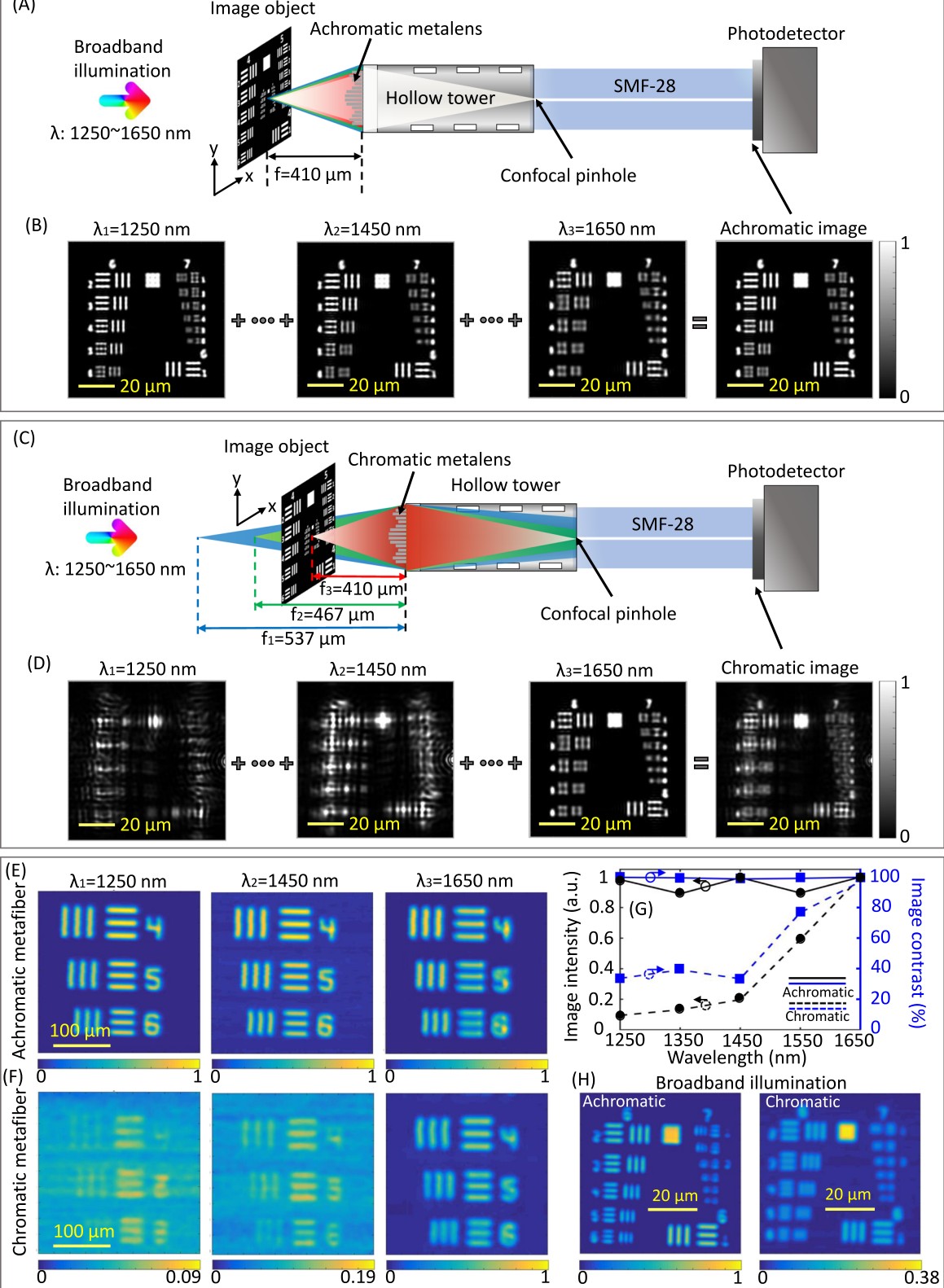

**Fig. 5 Imaging performance of an achromatic metafiber. A–D** Simulated confocal imaging results of the 1951 USAF resolution test chart (Groups 6 and 7) under broadband illumination, based on achromatic (**A**, **B**) and chromatic (**C**, **D**) metalenses-implemented SMFs, respectively. **E**, **F** Experimentally obtained confocal imaging results of the USAF resolution test chart (Group 5, Elements 4–6) based on the **E** achromatic and **F** chromatic metafibers at different wavelengths. **G** Averaged image intensity (black color) and image contrast (blue color) of the experimentally obtained images at different wavelengths, based on the achromatic (solid lines) and chromatic (dashed lines) metafibers, respectively. **H** Experimentally obtained confocal imaging results based on the achromatic (left) and chromatic (right) metafibers under broadband illumination that consists of 8 equally spaced wavelength channels ranging from 1250 to 1650 nm. We used the Group 6 Element 5 to mark the spatial resolution of our achromatic metafibre to be 4.92 μm.

that is implemented on a fiber bundle or a multi-core fiber (Fig. S16). Moreover, our achromatic metafiber could be used for femtosecond laser-assisted therapy and surgery[55,56], through the achromatic focusing of femtosecond laser pulses with a typical bandwidth of several hundreds of nanometers. Here we would like to highlight that due to the comparatively large transverse extent of the metasurface structure (on the order of 100 μm), the local light fluence in commonly used experimental configurations (e.g., nonlinear frequency conversion of optical trapping) is sufficiently low to prevent damage to the nanoprinted structure (Supplementary Note 9 and Fig. S17).

Alternatively, multimode fibers have been developed for endoscopic imaging[57–60], although a calibration setup is typically required to measure the fiber's transmission matrix by using a spatial light modulator and an interferometer, which increases the complexity of a fiber endoscope as opposed to the use of a single-mode fiber. In addition, the use of a single-mode delivery fiber removes the susceptibility of the fiber to external influences such as bending, which is useful for remote applications such as in vivo optical coherence tomography and optical sensing[6,42]. Therefore, our demonstrated highly compact and flexible achromatic metafiber unlocks the full potential of fiber meta-optics for widespread photonic applications, ranging from nonlinear fiber lasers and wavelength-multiplexing-based fiber-optic communications, to deep tissue imaging[61] and hyperspectral imaging in confocal endomicroscopy[62].

## Methods

**Numerical calculation of 3D meta-atoms**. Numerical simulation of phase, group delay, and conversion efficiency of 3D meta-atoms was carried out using our in-house developed rigorous coupled-wave analysis (RCWA). The scattering parameters (S-parameters) of a polymer-based meta-atom unit cell with periodic boundary conditions for the polarization along the long transverse axis $t_l$ and the short transverse axis $t_S$ were obtained from the zeroth-order transmission coefficient. According to the analytical results based on the Jones matrix calculus from our previous study[20], conversion efficiency and phase of cross-polarization term were defined as $CE = \left| \frac{t_l - t_s}{2} \right|^2$ and $\varphi = \tan^{-1}\left( \frac{\text{imag}\left(\frac{t_l - t_s}{2}\right)}{\text{real}\left(\frac{t_l - t_s}{2}\right)} \right)$, respectively. The corresponded group delay at the wavelengths of interest was calculated as $\frac{d\varphi}{d\omega} = \frac{\left( \varphi|_{\omega = \omega_{\min}} - \varphi|_{\omega = \omega_{\max}} \right)}{\omega_{\min} - \omega_{\max}}$, where $\omega_{\min,\max}$ represents the minimum and maximum frequencies in the regime of interest.

**3D laser nanoprinting of an achromatic metafiber**. 3D laser nanoprinting of the polymer-based achromatic metalens on-fiber was realized by two-photon polymerization via a tightly focused femtosecond laser beam. As such, this was accomplished by using a commercial photolithography system (Photonic Professional GT, Nanoscribe GmbH). The polymer metasurface samples were first fabricated on a silica substrate in IP-L 780 photoresist resin (Nanoscribe GmbH) by means of a high numerical aperture objective (Plan-Apochromat 63x/1.40 Oil DIC, Zeiss) in the dip-in configuration. The optimized printing parameters were 47.5 mW and 7000 μm/s for laser power and scanning speed, respectively. After laser exposure, the samples were developed by immersion in propylene glycol monomethyl ether acetate (PGMEA, Sigma-Aldrich) for 20 min, Isopropanol (IPA, Sigma-Aldrich) for 5 min and Methoxy-nonafluorobutane (Novec 7100 Engineered Fluid, 3 M) for 2 min. Finally, the fabricated samples were dried in air by evaporation. To increase the mechanical strength of polymer nanopillars with considerably high aspect ratios, a small hatching (laser movement step in the transverse directions: 20 nm) and slicing (laser movement step in the longitudinal direction: 50 nm) distances were employed in our 3D nanoprinting. To significantly increase the fabrication throughput, galvanic mirror scanning mode was selected and each scanning field was limited to a square unit cell of 100 μm by 100 μm to reduce stitching errors and optical aberrations.

In order to interface an achromatic metalens with a SMF, the 3D physical positions (x, y, and z coordinates) of a surface-cleaved SMF were first determined by an air objective (Plan-Apochromat 20x/0.85, Zeiss). Thereafter, the IP-L 780 photoresist resin was dropped onto the SMF and the translational stage went to predetermined 3D coordinates under a high numerical aperture objective (Plan-Apochromat 63x/1.40 Oil DIC, Zeiss). After finding the fiber end face under the high numerical aperture objective, the 3D laser nanoprinting of a hollow tower structure followed by an achromatic metalens on top of the tower was initiated. On the 3D-nanoprinted hollow tower, we created some rectangular holes on the side wall, allowing the inside photoresist intact from the laser exposure to be fully removed from the chemical development process. We printed a thin spacer layer

(height of 15 μm) between the 3D achromatic metalens and the tower structure, which creates a flat and smooth surface on top of the tower structure. We show that such a thin spacer layer induces only about 0–3% transmittance loss for the incidence angle up to 10 degrees corresponding to the maximal convergence angle of our achromatic metalens with NA = 0.21 (Fig. S18). The thin polymer film introduces Fabry-Perot resonances that lead to a noticeable transmittance modulation when the incidence angle is greater than 20 degrees, which does not fall into our case. In addition, we numerically show that introducing this thin spacer layer has negligible influence on the lens focusing (Fig. S19).

## Data availability
The data that support the findings of this study are available from the corresponding author upon request.

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

## Acknowledgements

H.R. acknowledges the funding support from Humboldt Research Fellowship from the Alexander von Humboldt Foundation and the DECRA Project (DE220101085) from the Australian Research Council. S.A.M. acknowledges the funding support from the Deutsche Forschungsgemeinschaft (MA 4699/2-1, MA 4699/7-1), the EPSRC (EP/M000044/1), and the Lee-Lucas Chair in Physics. M.A.S. acknowledges funding from the Deutsche Forschungsgemeinschaft via the grants SCHM2655/8-1, SCHM2655/11-1, SCHM2655/15-1, and SCH2655/21-1. J.R. acknowledges the POSCO-POSTECH-RIST Convergence Research Center program funded by POSCO, an industry-university strategic grant funded by LG Innotek, and the National Research Foundation (NRF) grants (NRF-2022M3C1A3081312, NRF-2022M3H4A1A02074314, CAMM-2019M3A6B3030637, NRF-2019R1A5A8080290) funded by the Ministry of Science and ICT of the Korean government. The Korea-Germany collaboration part is supported by the GEnKO program (NRF-2021K2A9A2A15000174) funded by the NRF. J.J. acknowledges the Hyundai Motor *Chung Mong-Koo* fellowship, and the NRF-DAAD Summer Institute program funded by the NRF and German Academic Exchange Service (DAAD). C.L. acknowledges the scholarship support from the China Scholarship Council. The authors acknowledge Dr. Jungho Mun (POSTECH) for the in-house RCWA code development and the implementation on construction of the meta-atom library. The authors acknowledge Dr. Jiawen Li (The University of Adelaide) for the technical advice on endoscope scanning head.

## Author contributions

H.R., M.A.S., and S.A.M. proposed the idea and conceived the experiment; J.J., J.R., and H.R. performed the theoretical and numerical simulations; C.L., and H.R. performed the 3D laser nanoprinting and device characterization; A.A. contributed to the tower design; M.P., J.K., and M.A.S. contributed to the fiber wavefront correction and supported on the experimental characterization. H.R., J.R., M.A.S., and S.A.M. contributed to the data analysis; H.R., J.J., M.A.S., J.R., and S.A.M. contributed to the paper revision; all the authors completed the writing of the paper.

## Competing interests

The authors declare no competing interests.
