## [Peer Review File · Nature Communications]

An achromatic metafiber for focusing and imaging across the entire telecommunication rangeREVIEWER COMMENTS

Reviewer #1 (Remarks to the Author):

In their article titled "an achromatic metafiber for focusing and imaging across the entire telecommunication range", the authors describe the design and the fabrication of a diffractive meta-lens 3d printed at the tip of a single mode fiber. The novelty of the meta-lens is that it is diffraction limited and achromatic over a 400 nm range between 1250 nm and 1650 nm. The on-axis performance of the meta lens is quite impressive over that wavelength range.

The field of microendoscopy with optical fibers is quite vast and has been researched a lot in the last decade. There should be additional references to the field as the manuscript is about imaging through optical fibers.

In particular, adding micro-optics by two photon printing on top of fibers was done first in this paper:

"Micro-Optics Fabrication on Top of Optical Fibers Using Two-Photon Lithography" IEEE PHOTONICS TECHNOLOGY LETTERS, VOL. 22, NO. 7, APRIL 1, 2010

Although the authors only reference metal-lenses research, the performance of the proposed metafiber endoscope imaging should be compared to imaging with other fiber micro-endoscope technology.

For example, the authors cite a 2019 article by Shin et al on a lens-less microendoscope. There have been multiple fiber lensless micro-endoscope demonstrations before that:

e.g

Choi, Y. et al "Scanner-Free and Wide-Field Endoscopic Imaging by Using a Single Multimode Optical Fiber," Physical Review Letters 109(12/ ...), 12285–12292 (2012).

Papadopoulos I. et al., "High-resolution, lensless endoscope based on digital scanning through a multimode optical fiber", Biomed. Opt. Express, Vol. 4, Issue 2, pp. 260-270, 2013.

E. R. Andresen, et al "Two-photon lensless endoscope," Opt. Express 21(18), 20713–20721 (2013)

.

D Loterie et al, "Confocal microscopy through a multimode fiber using optical correlation", Optics letters 40 (24), 5754-5757.

More recently, for wide field imaging, work on aberration correction in GRIN micro-lenses:

Antonini, Sattin, et al. « Extended field-of-view microendoscopy through aberration corrected GRIN lenses, " eLife 2020;9:e58882. DOI: <https://doi.org/10.7554/eLife.58882>

My main questions are on the imaging part with the meta lens on top of the fiber. On Fig 5. How are the images obtained: it was not clear in the text. Was the object moved in xy ?

If so, the metalens is only demonstrated as an on axis single focal point. Could the author elaborate on how this device can be used for imaging from the same side ?. In practical applications, an endoscope does not have access to the distal side to bring light of move the object.

For example, the metalens could be used with a fiber bundle of single mode fibers (same as for reference 13, 14). In this case, there will be off-axis imaging. How is the lens performing in off-axis ?

Reviewer #2 (Remarks to the Author):

Equation Chapter 1 Section 1 The authors propose, design, fabricate, and characterize an achromatic metalens interfacing with a commonly used single-mode fiber. The metalens is designed to perform achromatic and polarization insensitive focusing across the entire telecommunication wavelength band of the fiber, ranging from 1.25 to 1.65 μm . The proposed meta-atom takes the shape of rectangular nanopillar, which can be analyzed as a truncated waveguide. Two degrees of freedom is provided in the meta-atom design: the transverse dimensions, which controls the effective refractive indices; and height. The two degrees of freedom provide enough control for the phase and group delay profile at each radial position. The authors also proposed a polarization insensitive design by looking at the cross-polarization response of the birefringent meta-atom rotated at 0 and 90 degrees. The proposed metalens design was first characterized on planar substrate and then integrated on the end of a single-mode fiber. The fabrication process and experimental setup of both cases are discussed in detail. Excellent experimental performances are shown in the achromatic focusing response and broadband metafiber imaging. The manuscript includes detailed analytical design method as well as practical implementation.

In addition, please see our detailed comments below.

Q.1: Please comment on how the pitch distance P of the meta-atom was chosen, considering the meta-atom dimensions and the potential near-field coupling between adjacent meta-atoms. What is the minimal separation distance between various meta-atoms? In addition, the effective refraction index of each meta-atom design is simulated with a periodic boundary condition, which could be inaccurate when there is an abrupt change between the phase response of the adjacent meta-atom. What are the considerations designing the spatial sampling of the metalens' phase profile? Is the value of P kept constant throughout the surface? Or is it designed in such a way that P is smaller where the spatial phase gradient is large?

Q.2: Page 11, line 251. The authors state that "a small shift of focal position ($f=225 \mu\text{m}$) with respect to the nominal design ($f=250 \mu\text{m}$) at a wavelength of 1.65 μm was observed, which is due to the spatial sampling of a lens phase profile." Please elaborate more on the relationship between the spatial sampling and focal position. In addition, does the choice of pitch distance P affect the efficiency of the metalens?

Q.3: Please comment on the potential reasons for the low efficiency of the designed metalens. In the Supplementary Table 3, both the transmission efficiency and focusing efficiency show frequency-dependent behavior. That is, the efficiency of the designed metalens decreases notably with increasing frequency. Please elaborate the reason governing the frequency-dependent response.

Q.4: The polarization-insensitivity is enabled in this paper by introducing polarization loss. That is, only the cross-polarization phase and group delay is considered in the design. This limits the efficiency of the proposed metalens, and leads to an additional amplitude modulation across the surface. In general, a flat amplitude response is required for achromatic focusing metalens design. On page 10, line 240, the authors state that "The negligible deviation of the measured FWHM values with respect to the diffraction-limited ones might result from the additional amplitude modulation in the fabricated metalens." Please show a figure of the amplitude profile of the designed metalens, and elaborate more on its effect on the focusing performance.

Q.5: What limits the maximum height difference between different nanopillar meta-atom design? From Eq. (2), it seems that a simple waveguide treatment is used for analyzing the metalens, where the diffraction at the interface between the waveguide and air is neglected. This design approach could potentially lead to bad angular performance. For the proposed metalens design, does the height difference between individual meta-atoms limit the choice of the focal length? Can the authors provide more details on how to miniaturize the focal length of such metalens?

Q.6: Page 11, line 259: Please provide the design considerations for the hollow tower structure. Can the height of the hollow tower structure be further reduced? The diameter of the hollow tower

structure is 120 μm , whereas in Supplementary Note 3, the mode field diameter is 100 μm . Are the two values different for a reason?

Reviewer #3 (Remarks to the Author):

The authors presented a nanoprinted metalens attached to the end of a fiber to perform imaging. The lens is achromatic across the telecom band, and was demonstrated for broadband confocal imaging. The superior performance is ascribed to the freedom to control the height of nano unit cells by 3D printing.

κ in Eq. 1 is said to be unitless. As far as I can see, its unit should be radian.

In Eqs 1 and 2, is the dispersion of each individual 3D nanopillar present and accounted for? It appears that the authors assume n_{eff} to be constant across the bandwidth.

I was wondering if it's not possible to make a conventional convex achromatic lens at a fiber tip from a non-dispersive material.

The efficiency of the metalens was well quantified. Can the authors explain the sources of loss? Where did the remaining energy go?

Can this lens handle large laser power? Can the authors quantify the damage threshold?

In Fig. 1, what are those features on the side of the 3D-nanoprinted tower?

P. 13 refers to Fig. 12 that doesn't exist. Please fix.

Reviewer #1 (Remarks to the Author):

1. In their article titled “an achromatic metafiber for focusing and imaging across the entire telecommunication range”, the authors describe the design and the fabrication of a diffractive meta-lens 3d printed at the tip of a single mode fiber. The novelty of the meta-lens is that it is diffraction limited and achromatic over a 400 nm range between 1250 nm and 1650 nm. The on-axis performance of the meta lens is quite impressive over that wavelength range.

Reply 1: We thank the Reviewer for the endorsement of the novelty and high performance of our designed and fabricated achromatic metalens on a fiber.

2. The field of microendoscopy with optical fibers is quite vast and has been researched a lot in the last decade. There should be additional references to the field as the manuscript is about imaging through optical fibers. In particular, adding micro-optics by two photon printing on top of fibers was done first in this paper: “Micro-Optics Fabrication on Top of Optical Fibers Using Two-Photon Lithography” IEEE PHOTONICS TECHNOLOGY LETTERS, VOL. 22, NO. 7, APRIL 1, 2010

Reply 2: We agree with the Reviewer that the mentioned paper in the field of microendoscopy is highly relevant to our manuscript. We have added it as Ref. 44.

3. Although the authors only reference meta-lenses research, the performance of the proposed metafiber endoscope imaging should be compared to imaging with other fiber micro-endoscope technology. For example, the authors cite a 2019 article by Shin et al on a lens-less microendoscope. There have been multiple fiber lensless micro-endoscope demonstrations before that:

- e.g Choi, Y. et al “Scanner-Free and Wide-Field Endoscopic Imaging by Using a Single Multimode Optical Fiber,” Physical Review Letters 109(12/ ...), 12285–12292 (2012).
- Papadopoulos I. et al., “High-resolution, lensless endoscope based on digital scanning through a multimode optical fiber”, Biomed. Opt. Express, Vol. 4, Issue 2, pp. 260-270, 2013.
- E. R. Andresen, et al “Two-photon lensless endoscope,” Opt. Express 21(18), 20713–20721 (2013) .
- D Loterie et al, “Confocal microscopy through a multimode fiber using optical correlation”, Optics letters 40 (24), 5754-5757.

More recently, for wide field imaging, work on aberration correction in GRIN micro-lenses:

- Antonini, Sattin, et al. « Extended field-of-view microendoscopy through aberration corrected GRIN lenses, “ eLife 2020;9:e58882. DOI: <https://doi.org/10.7554/eLife.58882>

Reply 3: We thank the Reviewer for a careful survey of literature. We agree with the Reviewer that our achromatic metafiber should also be compared with other fiber micro-endoscope technology. We have added the work in GRIN micro-lens based on two-photon lithography as Ref. 45. We have also added one sentence to compare our work based on a single-mode fiber with the suggested references (Refs. 51-53) using multimode fibers:

Page 17 in the main text:

“Alternatively, multimode fibers have been developed for endoscopic imaging⁵⁷⁻⁶⁰, although a calibration setup is typically required to measure the fiber’s transmission matrix by using a spatial light modulator and an interferometer, which increases the complexity of a fiber endoscope as opposed to the use of a single-mode fiber.”

4. My main questions are on the imaging part with the meta lens on top of the fiber. On Fig 5. How are the images obtained: it was not clear in the text. Was the object moved in xy ? If so, the metalens is only demonstrated as an on axis single focal point. Could the author elaborate on how this device can be used for imaging from the same side ? In practical applications, an endoscope does not have access to the distal side to bring light of move the object.

For example, the metalens could be used with a fiber bundle of single mode fibers (same as for reference 13, 14). In this case, there will be off-axis imaging. How is the lens performing in off-axis ?

Reply 4: We thank the Reviewer for insightful thoughts and professional comments.

Yes, in our proof-of-concept experiment (Fig. 5), we fixed the achromatic metafiber position and scanned the object in the transverse directions by using a translational stage. Indeed, we mentioned how the images were collected in our experiment (Page 13): “Transmitted light under a broad illumination from a moving object placed on a 3D piezo nanopositioning stage was directly collected by the achromatic metafiber (the metalens side).”

Our current paper focuses on a proof-of-concept that achromatic meta-optics can be implemented on a single-mode fiber for scanning imaging, and we do not claim that the present fiber can be used for endoscopy. We agree with the Reviewer that a practical endoscope does not have access to the distal side light illumination and movement of the object. To employ our achromatic metafiber for practical endoscopic imaging, we will need to integrate the achromatic metafiber with a distal-scanning endoscope⁵², as well as the use of a fiber-based beamsplitter to separate the excitation and fluorescence signals from a sample, as shown in Fig. S14. Alternatively, if the excitation wavelength is beyond the single-mode fiber bandwidth, a micro-structured fiber could be used⁵³, where the excitation and collection light signals are transported through individual guiding domains. To elaborate the advantages of using telecommunication wavelengths for deep-tissue imaging, as well as the possible implementation of our achromatic metalens on a distal-scanning endoscope, we

have added a few sentences in the main text and a Supplementary Figure 14. We will regard this implementation that is related to endoscopic application as our future work.

Page 15 in the main text:

“In addition, although the telecommunication wavelength range was typically avoided due to the presence of a water overtone absorbance peak, it allows deep tissue imaging with large penetration depths⁵¹. This longer near-infrared wavelength range has facilitated higher imaging clarity, due to its exponentially reduced photon scattering and autofluorescence⁵¹. As such, we envision that our achromatic metafiber can be fabricated on a distal-scanning endoscope for practical endoscopic imaging⁵², in which a fiber-based beamsplitter can be used to separate the excitation and fluorescence signals from a fluorescence sample (Fig. S14). Alternatively, if the excitation wavelength is beyond the SMF bandwidth, a micro-structured fiber can be used⁵³, where the excitation and collection light signals are transported through individual guiding domains.”

Figure S14. Suggestion for an achromatic metafiber on a forward-viewing endoscope for practical endoscopic imaging. A fiber-based beamsplitter is used to separate fluorescence signal from the laser excitation. A forward-viewing endoscope integrates an achromatic metafiber with a piezoelectric transducer (PZT) tube for distal-end confocal scanning imaging in the focal region. The achromatic metafiber can simultaneously excite and collect broadband fluorescence signals for practical endoscopic imaging.

To verify the achromatic metalens performance for off-axis imaging, we have conducted a 3D Finite-Difference Time-Domain (FDTD) simulation using our selected meta-atoms for implementing the desired phase and group delay responses in Figs. 2E and 2F, respectively. We characterized the focusing performance of the achromatic metalens based on different incident angles at different wavelengths (Fig. S15). Specifically, we simulated the off-axis focusing performance based on an oblique incidence angle up to ± 7.5 degrees, which correspond to the lateral shifts of $\pm 52 \mu\text{m}$ (@1650 nm) in the focal plane (Figs. S15A and S15B). We simulated the focusing performance at three wavelengths of 1250 nm, 1450 nm and 1650 nm. Our results show that our achromatic metalens can also compensate the chromatic aberration for off-axis incident beams for the incident angle up to ± 7.5 degrees. The off-axis focusing efficiency keeps flat for an oblique incidence angle up to 5 degrees (Fig. S15C). As such, our simulation results suggest that our designed achromatic metalens (without further

optimization) could exhibit a good performance for off-axis imaging up to ± 7.5 degrees, which opens the possibility of implementing it on a multicore fiber or a fiber bundle for the wide-field imaging with a large field-of-view of $104 \mu\text{m}$ (@1650 nm).

Figure. S15. Numerical characterization of off-axis focusing of an achromatic metalens. (A) Schematic of a 3D FDTD model used for studying the performance of off-axis focusing of an achromatic metalens on a glass substrate. **(B)** Intensity distributions of the metalens focus in the longitudinal focal plane based on the incident angles of 2° , 4° and 7.5° at a wavelength of 1650 nm . Characterization of **(C)** the focal length and **(D)** focusing efficiency of the achromatic metalens under different incident angles at three wavelengths of 1250 nm , 1450 nm and 1650 nm .

To achieve wide-field imaging, we can implement our achromatic metalens with high-performing on- and off-axis imaging capability on top of a fiber bundle or a multicore fiber that consists of multiple SMFs (Fig. S16). We must mention, however, there is a general technical challenge that may limit the imaging quality of an achromatic metafiber bundle. Since the outputs of SMFs in the fiber bundle are all faced perpendicular to the fiber surface, the collection efficiency for off-axis imaging through the out-ring SMFs should be reduced as compared to the on-axis case (Fig. S16). The collected off-axis imaging signal by our achromatic metalens has an angle offset β with respect to the nominal fiber emission/collection direction (solid lines in Fig. S16). This angle mismatch could reduce the

collection (fiber coupling) efficiency for off-axis light signals and hence limit the field-of-view of an achromatic metalens. We believe this technical challenge is intrinsic to the use of a fiber bundle or a multicore fiber for wide-field imaging, which should also apply to the Reviewer mentioned Refs. 13 and 14.

Figure S16. Schematic of an implementation of an achromatic metalens on top of a multicore fiber for wide-field imaging. The light paths of on- (dashed lines) and off-axis (solid lines) imaging are shown. The off-axis light paths induce an angle offset β with respect to the nominal emission/collection direction of the SMF, which can reduce the collection efficiency of off-axis light signals and limit the field-of-view of the metalens.

To clarify this point, we have added several sentences in the main text and a Supplementary Note 8, Supplementary Figures S15 and S16.

Page 16 in the main text:

“Apart from high-performing on-axis imaging, our demonstrated achromatic metalens also exhibits a good performance for off-axis imaging up to ± 7.5 degrees that correspond to a maximal field of view of $104 \mu\text{m}$ (@1650 nm) in the focal plane (Fig. S15). We believe, however, there is a general technical challenge in the fiber-based wide-field imaging, as the off-axis light path could induce an angular offset with respect to the fiber axis (the nominal emission and collection direction of a SMF). This could reduce the collection efficiency of off-axis imaging and limit the field-of-view of the achromatic metalens that is implemented on a fiber bundle or a multi-core fiber (Fig. S16).”

“Supplementary Note 8. Characterization of off-axis focusing of an achromatic metalens.

To verify the achromatic metalens performance for off-axis imaging, we have conducted a 3D Finite-Difference Time-Domain (FDTD) simulation using our selected meta-atoms for

implementing the desired phase and group delay responses in Figs. 2E and 2F, respectively. We characterized the focusing performance of the achromatic metalens based on different incident angles at different wavelengths (Fig. S15). Specifically, we simulated the off-axis focusing performance based on an oblique incidence angle up to ± 7.5 degrees, which correspond to the lateral shifts of $\pm 52 \mu\text{m}$ (@1650 nm) in the focal plane (Figs. S15A and S15B). We simulated the focusing performance at three wavelengths of 1250 nm, 1450 nm and 1650 nm. Our results show that our achromatic metalens can also compensate the chromatic aberration for off-axis incident beams for the incident angle up to ± 7.5 degrees. The off-axis focusing efficiency keeps flat for an oblique incidence angle up to 5 degrees (Fig. S15C). As such, our simulation results suggest that our designed achromatic metalens (without further optimization) could exhibit a good performance for off-axis imaging up to ± 7.5 degrees, which opens the possibility of implementing it on a multicore fiber or a fiber bundle for the wide-field imaging with a large field-of-view of $104 \mu\text{m}$ (@1650 nm).

To achieve wide-field imaging, we can implement our achromatic metalens with high-performing on- and off-axis imaging capability on top of a multicore fiber that consists of multiple SMFs (Fig. S16). We must mention, however, there is a general technical challenge that may limit the imaging quality of the achromatic metalens that is implemented on a multicore fiber (or a fiber bundle). Since the outputs of SMFs in the multicore fiber are all faced perpendicular to the fiber surface, the collection efficiency for off-axis imaging through the out-ring SMFs should be reduced as compared to the on-axis case (Fig. S16). The collected off-axis imaging signal by our achromatic metalens has an angle offset with respect to the nominal fiber emission/collection direction (solid lines in Fig. S16). This angle mismatch could reduce the collection (fiber coupling) efficiency for off-axis light signals and hence limit the field-of-view of an achromatic metalens, although the use of a multicore fiber could mitigate this mismatch. We believe this technical challenge is intrinsic to the use of a multicore fiber or a fiber bundle for wide-field imaging, which should also apply to Refs. 13 and 14.”

Reviewer #2 (Remarks to the Author):

1. The authors propose, design, fabricate, and characterize an achromatic metalens interfacing with a commonly used single-mode fiber. The metalens is designed to perform achromatic and polarization insensitive focusing across the entire telecommunication wavelength band of the fiber, ranging from 1.25 to 1.65 μm . The proposed meta-atom takes the shape of rectangular nanopillar, which can be analyzed as a truncated waveguide. Two degrees of freedom is provided in the meta-atom design: the transverse dimensions, which controls the effective refractive indices; and height. The two degrees of freedom provide enough control for the phase and group delay profile at each radial position. The authors also proposed a polarization insensitive design by looking at the cross-polarization response of the birefringent meta-atom rotated at 0 and 90 degrees. The proposed metalens design was first characterized on planar substrate and then integrated on the end of a single-mode fiber. The fabrication process and experimental setup of both cases are discussed in detail. Excellent experimental performances are shown in the achromatic focusing response and broadband metafiber imaging. The manuscript includes detailed analytical design method as well as practical implementation.

Reply 1: We are very grateful to the Reviewer for the careful review of our manuscript and the very positive evaluation of our work.

2. In addition, please see our detailed comments below.

Q.1: Please comment on how the pitch distance P of the meta-atom was chosen, considering the meta-atom dimensions and the potential near-field coupling between adjacent meta-atoms. What is the minimal separation distance between various meta-atoms? In addition, the effective refraction index of each meta-atom design is simulated with a periodic boundary condition, which could be inaccurate when there is an abrupt change between the phase response of the adjacent meta-atom. What are the considerations designing the spatial sampling of the metalens' phase profile? Is the value of P kept constant throughout the surface? Or is it designed in such a way that P is smaller where the spatial phase gradient is large?

Reply 2.1: We thank the Reviewer for pointing out these questions that should be clarified for further improving the technical soundness of our paper. To determine the pitch distance of meta-atoms, we have considered the trade-off between the near-field coupling and the lens sampling rate. As a result, we set a constant pitch of 2.2 μm in the metalens through balancing 1) high cross-polarization conversion efficiency; 2) sufficient sampling of a lens function with high efficiency; 3) negligible near-field coupling. To clarify this important point, we have added a Supplementary Note 7:

“Supplementary Note 7. Meta-atom size and pitch considerations.

Here we kept a constant pitch distance for all meta-atoms throughout the metalens surface. The geometry of meta-atom was optimized to maximize its interaction with incident light. In general, the effective length (the product of refractive index and geometric length) of meta-atom should be comparable to the incident wavelength. Given the low refractive index of our used polymer meta-atoms, the required geometric length can be approximated as $1.1 \mu\text{m}$ ($\frac{1.65\mu\text{m} (\text{nominal wavelength})}{1.5 (\text{refractive index})}$). Given the fabrication constraints of two-photon laser lithography, we considered a duty cycle of 50% in a meta-atom unit cell, and therefore we used a pitch distance of $2.2 \mu\text{m}$ in our paper. It should be mentioned that under this pitch distance, meta-atoms with different in-plane aspect ratios exhibit high cross-polarization conversion efficiency (SI Fig. 1), which is crucial for designing an achromatic metalens of high efficiency.

SI Fig. 1. Polarization conversion efficiency of 3D meta-atoms with fixed in-plane aspect ratios of 0.4 (A), 0.5 (B) and 0.6 (C) at a nominal wavelength of $1.65 \mu\text{m}$. The pitch distance of meta-atoms was fixed at $2.2 \mu\text{m}$.

Second, the near-field coupling between adjacent meta-atoms should also be considered. To clarify this point, we simulated effective refractive indices of the transverse electric (TE) and transverse magnetic (TM) modes of the 2D cross-sections of 3D nanopillars (inset figure) across the wavelength of interests, wherein the pitch of nanopillar meta-atom was varied from $1.7 \mu\text{m}$ to $10 \mu\text{m}$ (SI Fig. 2). We simulated the effective refractive indices of a meta-atom with a length of $1.625 \mu\text{m}$ and a width of $0.813 \mu\text{m}$. In particular, the effective refractive indices are sharply increased when the pitch gets smaller than $2 \mu\text{m}$, indicating modal hybridization and thus a strong near-field coupling between adjacent meta-atoms. On the other hand, meta-atoms with a pitch of above $3 \mu\text{m}$ show a nearly constant effective refractive index and thereby a negligible near-field coupling between adjacent meta-atoms. However, the increase of pitch will reduce the metalens efficiency, due to the Bragg diffraction in higher diffraction orders and insufficient sampling of a lens profile.”

SI Fig. 2. Effective refractive indices of TE and TM modes of a 3D nanopillar across the wavelength of interest. The pitch distance was varied from 1.7 μm to 10 μm. The nanopillar transverse dimensions of length and width were fixed as 1.625 μm and 0.813 μm, respectively.

Q.2: Page 11, line 251. The authors state that “a small shift of focal position ($f=225\ \mu\text{m}$) with respect to the nominal design ($f=250\ \mu\text{m}$) at a wavelength of $1.65\ \mu\text{m}$ was observed, which is due to the spatial sampling of a lens phase profile.” Please elaborate more on the relationship between the spatial sampling and focal position. In addition, does the choice of pitch distance P affect the efficiency of the metalens?

Reply 2.2: We have clarified in the above Reply on how to determine the pitch distance P . Here we focus on the relationship between the spatial sampling and focal position/focusing efficiency. To clarify this important point, we have added some discussion in the main text and a Supplementary Note 7:

Page 16 in the main text:

“In addition, to determine the pitch distance of meta-atoms, we have considered the trade-off between the near-field coupling and the lens sampling rate. We set a constant pitch of $2.2\ \mu\text{m}$ in the metalens through balancing i) high cross-polarization conversion efficiency; ii)

sufficient sampling of a lens function with high efficiency; iii) negligible near-field coupling (Supplementary Note 7).”

Supplementary Note 7. Meta-atom size and pitch considerations.

“Third, we considered the relationship between the spatial sampling and focal position/focusing efficiency. Indeed, insufficient sampling causes incorrect digitalization of a lens phase function, and therefore, creating some side effects such as a focal shift and decreased focusing efficiency. For a better comparison, we further simulated the performance of a same metalens profile under different sampling conditions, using diffraction theory based on the Fraunhofer approximation (given the fact that the NA of our metalens is small). Different sampling rates (corresponding to the metalens pitch distances) were chosen to investigate their influence on the lens performance, including focal length and focusing efficiency (SI Table 2). The focusing efficiency is defined as the integrated intensity over three times of the full width at half maximum (FWHM) area in the focal plane with respect to the integrated intensity over the whole lens aperture.

Our results indicate that a large pitch used for digitizing the lens function (with a NA of 0.2 at the wavelength of 1.65 μm , a lens diameter and a focal length of 100 μm and 240 μm , respectively) can reduce the lensing performance by shifting the focal length and reducing the lens focusing efficiency, which becomes obvious when the pitch distance is larger than 1 μm . We set a constant pitch distance of 2.2 μm in our work through balancing 1) high cross-polarization conversion efficiency; 2) sufficient sampling of a lens function with a relatively high efficiency; 3) negligible near-field coupling. Fabrication of an achromatic metalens with a pitch below 1 μm would be very challenging due to the large aspect ratio between the height and transverse dimensions, which reaches up to 33.75 in our work. Moreover, according to our simulation in SI Table 2, the pitch smaller than 1 μm can incur pronounced near-field coupling between nanopillars.

SI Table 2. Simulated focal length and focusing efficiency of a metalens function by varying the pitch distance and pixel number of the lens with a fixed numerical aperture of 0.2 (a diameter of 100 μm and a desired focal position of 240 μm). This is equivalent to our metalens on glass designed in Fig. 2. The wavelength was fixed as 1650 nm. The used pitch distance by this work is in bold.

Pitch distance (μm)	Focal length (μm)	Focusing efficiency
4	218	0.505
3	223	0.529
2.2	229	0.607
2	231	0.615
1	236	0.747
0.5	238	0.824
0.2	239	0.886

0.1	240	0.920
-----	-----	-------

”

Q.3: Please comment on the potential reasons for the low efficiency of the designed metalens. In the Supplementary Table 3, both the transmission efficiency and focusing efficiency show frequency-dependent behavior. That is, the efficiency of the designed metalens decreases notably with increasing frequency. Please elaborate the reason governing the frequency-dependent response.

Reply 2.3: The frequency-dependent efficiency is attributed to the fact that the longest wavelength (1.65 μm) was set for nominal metalens design. Meta-atom dimensions were numerically optimized to exhibit strong polarization conversion efficiency particularly at this nominal wavelength, exhibiting higher conversion efficiency than other wavelengths (Fig. 2B). This is the major reason why the metalens efficiency is frequency-dependent and decreases by increasing frequency (Figs. 3B, 3E and 4E).

To clarify this point, we have added one sentence in the main text:

Page 10 in the main text:

“We have to mention that our meta-atoms were numerically optimized to exhibit strong polarization conversion efficiency particularly at the nominal wavelength of 1650 nm for the metalens design, exhibiting higher conversion efficiency than other wavelengths (Fig. 2B). As such, the metalens efficiency appears to be frequency-dependent and decreases by increasing frequency (Fig. 3B).”

Q.4: The polarization-insensitivity is enabled in this paper by introducing polarization loss. That is, only the cross-polarization phase and group delay is considered in the design. This limits the efficiency of the proposed metalens, and leads to an additional amplitude modulation across the surface. In general, a flat amplitude response is required for achromatic focusing metalens design. On page 10, line 240, the authors state that “The negligible deviation of the measured FWHM values with respect to the diffraction-limited ones might result from the additional amplitude modulation in the fabricated metalens.” Please show a figure of the amplitude profile of the designed metalens, and elaborate more on its effect on the focusing performance.

Reply 2.4: We thank the Reviewer for the professional comment. We agree with the Reviewer that an additional non-uniform amplitude distribution in the cross-polarization response may deviate the metalens performance from the diffraction-limited focusing. To quantify this effect, we have added a simulation based on diffraction theory. We compared two cases of a lens function (used by our metalens) with and without a non-uniform amplitude extracted from our designed metalens (shown in Fig. 2E) in Fig. S9. We have also provided a quantitative comparison of the focusing performance between the uniform and non-uniform amplitude

profiles in Fig. S9. It indicates that the non-uniform amplitude profile does not degrade the diffraction-limited focus, which is evidenced by the fact the lens with a non-uniform amplitude exhibits a same FWHM, a similar focal length and a slightly higher focusing efficiency as compared to the case of the lens with a uniform amplitude.

To clarify this point, we have added a few sentences in the main text and a Supplementary Fig. 9:

Page 11 in the main text:

“We evaluated how an additional non-uniform amplitude distribution in the cross-polarization component affects the diffraction-limited focusing of our metalens. We simulated and compared two cases of a spherical lens function (used by our metalens) with and without a non-uniform amplitude distribution extracted from our designed metalens in Fig. 2E, the results are shown in Fig. S9. It indicates that the non-uniform amplitude does not degrade the diffraction-limited focus, exhibiting a same FWHM, a similar focal length and a slightly higher focusing efficiency (due to the complex-amplitude modulation) as compared to the case of a uniform amplitude profile.”

	Focal length (μm)	Focusing efficiency	FWHM (μm)
Uniform amplitude	228	0.607	4.604
Non-uniform amplitude	224	0.657	4.604

Fig. S9. Comparison of the lens performance between a uniform amplitude distribution and a non-uniform amplitude distribution extracted from metalens design in Figure 2 in the main text. (A) The phase profile of a spherical lens function (used by our metalens). (B) A uniform amplitude profile. (C) Meta-atoms-based non-uniform amplitude profile. (D and E) Intensity distributions of the lens function in the XZ focal plane with a uniform amplitude

profile (D) and with a non-uniform amplitude profile (E). (F and G) The counterparts of (D and E) showing the intensity distributions in the transverse XY focal plane. The table compares focusing performance of lenses with uniform and non-uniform amplitude profiles.

Q.5: What limits the maximum height difference between different nanopillar meta-atom design? From Eq. (2), it seems that a simple waveguide treatment is used for analyzing the metalens, where the diffraction at the interface between the waveguide and air is neglected. This design approach could potentially lead to bad angular performance. For the proposed metalens design, does the height difference between individual meta-atoms limit the choice of the focal length? Can the authors provide more details on how to miniaturize the focal length of such metalens?

Reply 2.5: We thank for the Reviewer's insightful comments. First, we would like to clarify that our metalens based on subwavelength nanopillar waveguides works in the metasurface regime, in which individual nanopillars work as independent scattering elements with judiciously controlled amplitude, phase, polarization and group delay responses. Even though the Bragg diffraction in principle exists in our used pitch distance of $2.2\ \mu\text{m}$ larger than the wavelength, we have carefully selected the nanopillars with negligible Bragg diffraction. To do so, nanopillars with high cross-polarisation conversion efficiency were chosen when we constructed the meta-atom library (Figure 2B). In this case, the diffraction at the interface between the waveguide and air exhibits only a negligible effect. Therefore, the height difference in meta-atoms should not limit the choices on the focal length and NA of an achromatic metalens with negligible Bragg diffraction. Moreover, to design a high-NA metalens, we can arrange 3D nanopillars with larger heights on the edge and smaller heights in the centre, so that the shadowing effect (light rays deviate from desired propagating paths due to the deflection at boundaries between the lens elements) can be avoided.

However, more considerations should be given when we design a high-NA achromatic metalens, as shown in Supplementary Fig. S5. The NA and bandwidth of an achromatic metalens form a trade-off in the achromatic metalens design, which is governed by the upper limit of time-bandwidth product (TBP) of meta-atoms, wherein TBP represents a product of the temporal duration ΔT and the spectral bandwidth $\Delta\omega$. It should be highlighted that our developed 3D achromatic metalens offers a largely extended TBP as compared to conventional metalenses (Fig. S1 and Table 1), due to the unleashed height degree of freedom in 3D nanopillar meta-atoms (Equations 1 and 2 in the main text). Owing to this expanded TBP limit, we have more flexible choices on the NA and bandwidth of an achromatic metalens, as shown in Fig. S5. Unlike the metalens design for a single wavelength, NA of the metalens can be increased by simply controlling geometry of the lens (enlarging the diameter or reducing the focal length). However, to increase the NA of an achromatic metalens, we have to consider the fundamental trade-off between the bandwidth and NA of an achromatic metalens based on the TBP consideration (Fig. S5).

To illustrate this important point, we have added a paragraph in the main text:

Page 8 in the main text:

“It should be mentioned that, owing to this expanded TBP limit by our 3D meta-atoms, we can have more flexible choices on the NA and bandwidth of an achromatic metalens. Unlike a metalens optimized for a single wavelength, the NA of which can be increased by either enlarging the lens diameter or reducing the focal length, here we have to consider the fundamental trade-off between the bandwidth and NA of an achromatic metalens based on the TBP consideration (Fig. S5B).”

Figure S5B. Theoretically calculated fundamental trade-off between the bandwidth and NA of an achromatic metalens based on the TBP consideration. The focal distance is fixed at $f = 250 \mu\text{m}$. The center wavelength was set to be 1450 nm. Black curves present the results based on our 3D meta-atom design library. Green dot: our current achromatic metalens design in this paper.

Q.6: Page 11, line 259: Please provide the design considerations for the hollow tower structure. Can the height of the hollow tower structure be further reduced? The diameter of the hollow tower structure is $120 \mu\text{m}$, whereas in Supplementary Note 3, the mode field diameter is $100 \mu\text{m}$. Are the two values different for a reason?

Reply 2.6: We thank the Reviewer for the professional comment. The hollow tower structure printed on top of a SMF was used for hosting the metalens, and the height of the hollow tower structure was determined from the expanded fiber output having a mode field diameter large enough to cover the whole metalens. To avoid the fiber beam hitting the side-wall of the tower, the hollow tower was designed to have a diameter slightly larger than the mode field diameter of the fiber output at the position of the metasurface. The height of the tower can be further reduced when a smaller metalens is designed. To clarify this point, we have added one sentence in the main text:

Page 12 in the main text:

“The height of the hollow tower structure was determined from the expanded fiber output having a mode field diameter large enough to cover the whole metalens. To avoid the fiber beam hitting the side-wall of the tower, the hollow tower was designed to have a diameter slightly larger than the mode field diameter of the fiber output at the position of the metalens.”

Reviewer #3 (Remarks to the Author):

1. The authors presented a nanoprinted metalens attached to the end of a fiber to perform imaging. The lens is achromatic across the telecom band, and was demonstrated for broadband confocal imaging. The superior performance is ascribed to the freedom to control the height of nano unit cells by 3D printing.

Reply 1: We thank the Reviewer for the very positive evaluation of our work.

2. κ in Eq. 1 is said to be unitless. As far as I can see, its unit should be radian.

Reply 2: We thank the Reviewer for pointing out. We agree that the unit of κ should be radian indeed, since the time-bandwidth product (TBP) was defined as

$$\text{TBP} = \Delta\omega\Delta T \leq \kappa,$$

where $\Delta\omega$ is the bandwidth of angular frequency in the unit of rad·Hz and ΔT is the temporal duration in the unit of second (1/Hz). Therefore, the unit of upper bound TBP κ in this paper should be radian, as the Reviewer pointed out. We have corrected this unit throughout the paper.

3. In Eqs 1 and 2, is the dispersion of each individual 3D nanopillar present and accounted for? It appears that the authors assume n_{eff} to be constant across the bandwidth.

Reply 3: We thank the Reviewer for insightful comments. Indeed, the dispersion of individual 3D nanopillar waveguides has been considered in Eqs. 1 and 2 in the main text. To clarify this important point, we have added a **Supplementary Note 2** to derive the TBP of an individual nanopillar waveguide.

“Supplementary Note 2. Derivation of the time-bandwidth product of a nanopillar waveguide.

For the light propagation in a waveguide with restricted transverse extension, the phase constant of the waveguide equals the wavenumber in the vacuum (k_0) times the effective refractive index (n_{eff}): $k = n_{\text{eff}}k_0 = n_{\text{eff}}\frac{\omega}{c}$, where ω is the angular frequency and c is the speed of light. The nanopillar waveguide can be regarded as a dielectric lossless slow-light device, its group velocity v_g can be derived as:

$$v_g = \frac{\partial\omega}{\partial k} = \frac{c}{n_{\text{eff}} + \omega \frac{dn_{\text{eff}}}{d\omega}}, \quad (3)$$

The group index $S(\omega)$ can be defined as:

$$S(\omega) = \frac{c}{v_g} = n_{eff} + \omega \frac{dn_{eff}}{d\omega}, \quad (4)$$

For a polymer nanopillar waveguide used in our interested spectral range, the dispersion of the effective refractive index is linear to the frequency (Fig. S2), Eq. 4 can be described as:

$$\frac{dn_{eff}}{d\omega} = \frac{n_{max} - n_{min}}{\omega_{max} - \omega_{min}}, \quad (5)$$

where n_{min} and n_{max} are the minimum and maximum effective refractive indices of the waveguide across the frequency range from ω_{min} to ω_{max} . Therefore, the group index at a central frequency (ω_0) can be rewritten as:

$$S(\omega_0) = n_{eff}(\omega_0) + \frac{\omega_0(n_{max} - n_{min})}{\Delta\omega}. \quad (6)$$

The group velocity of the ideal device can therefore be calculated as:

$$v_g = \frac{c}{S(\omega_0)} = \frac{c}{n_{eff}(\omega_0) + \frac{\omega_0(n_{max} - n_{min})}{\Delta\omega}}. \quad (7)$$

The time delay of the waveguide medium T with height H can be derived as:

$$\begin{aligned} T &= H \left(\frac{1}{v_g} - \frac{1}{c} \right) = H \left(\frac{n_{eff}(\omega_0) - 1 + \frac{\omega_0(n_{max} - n_{min})}{\Delta\omega}}{c} \right) \\ &= H \left(\frac{n_{eff}(\omega_0) - 1}{c} + \frac{\omega_0(n_{max} - n_{min})}{c\Delta\omega} \right). \end{aligned} \quad (8)$$

Therefore, the time-bandwidth product becomes

$$T\Delta\omega = H \left[\frac{n_{eff}(\omega_0) - 1}{c} (\omega_{max} - \omega_{min}) + \frac{(n_{max} - n_{min})}{c} \omega_0 \right]. \quad (9)$$

In general, the frequency bandwidth of a slow-light waveguide is much smaller than the central frequency: $\omega_{max} - \omega_{min} \leq \omega_0$, Eq. R8 can be further reduced to:

$$T\Delta\omega = \frac{\omega_0}{c} H(n_{max} - n_{min}), \quad (10)$$

which is consistent with the Eq. 1 result in the main text. Therefore, the time-bandwidth product in Eq. 10 (same as Eq. 1 in the main text) is mainly contributed from the dispersion of the effective refractive index of an individual nanopillar waveguide.”

4. I was wondering if it's not possible to make a conventional convex achromatic lens at a fiber tip from a non-dispersive material.

Reply 4: As far as we understand, there is a significant size mismatch between a conventional bulky achromatic doublet lens (made of different glass materials with higher (concave element) and lower (convex element) material dispersions) (Fig. R1A) and a tiny optical fiber end face (Fig. R1B). As an example, a commercial achromatic lens (e.g., a provider: Thorlabs.com) has a size of at least 200 times larger than the end surface of a single-mode fiber. Therefore, we reckon that it should be extremely challenging to directly interface such a bulky achromatic lens with a fiber tip.

Fig. R1. Size comparison of a conventional achromatic doublet lens and an achromatic metafiber on top of a single-mode fiber (SMF) with a diameter of 120 μm .

5. The efficiency of the metalens was well quantified. Can the authors explain the sources of loss? Where did the remaining energy go?

Reply 5: Indeed, there are two major sources of loss: (i) co-polarization component in meta-atoms; (ii) insufficient sampling of the lens profile. According to our description in Supplementary Note 6, the electric field passing through a single subwavelength birefringent nanopillar waveguide can be defined as:

$$\begin{bmatrix} \tilde{E}_x \\ \tilde{E}_y \end{bmatrix} = \frac{\tilde{\epsilon}_l + \tilde{\epsilon}_s}{2} \begin{bmatrix} 1 \\ \pm i \end{bmatrix} + \frac{\tilde{\epsilon}_l - \tilde{\epsilon}_s}{2} e^{\pm i2\alpha} \begin{bmatrix} 1 \\ \mp i \end{bmatrix}, \quad (26)$$

where \pm and \mp represent the co- and cross-polarization responses, respectively; $\tilde{E}_{x,y}$: output electric field along the Cartesian coordinates x and y -axis; $\tilde{t}_{l,s}$: complex-amplitude transmission coefficients of the meta-atom along short and long axis; α : in-plane rotation angle of a meta-atom. For the meta-atoms with asymmetric in-plane geometry, it can convert input circular polarization into the opposite helicity, for instance, left-handed circular polarization to right-handed circular polarization or vice versa. The cross-polarization term in Eq. 26 consists of a geometric phase (polarization-dependent) term $e^{\pm i2\alpha}$ that depends on the in-plane rotation angle of the meta-atom, as well as a polarization-independent term $\frac{\tilde{t}_l - \tilde{t}_s}{2}$ that contributes to the propagation phase. Based on Eq. 26, we can see that the co-polarization component does not contribute to the required geometric phase used for implementing the lens function at the nominal wavelength of 1650 nm. To design an achromatic metalens, we have sourced a large library of 3D meta-atoms to cover a broad span of group delay, some of meta-toms exhibit a small co-polarization component (we show a few examples in Table R1), which drops the metalens efficiency. As such, we attribute the co-polarisation component in 3D meta-atoms as one of the loss channels.

Table R1. Simulated cross- and co-polarization responses of selected meta-atoms.

	H (μm)	L (μm)	R	Cross-polarization			Co-polarization		
				1650 nm	1450 nm	1250 nm	1650 nm	1450 nm	1250 nm
Meta-atom I	11.75	1.3	0.3	0.90	0.88	0.46	0.06	0.03	0.04
Meta-atom II	10.75	1.575	0.3	0.88	0.74	0.50	0.07	0.02	0.26
Meta-atom III	13.25	1.5	0.5	0.55	0.60	0.17	0.06	0.23	0.25
Meta-atom IV	11	1.25	0.4	0.88	0.81	0.12	0.07	0.05	0.25

We attribute the other major source of loss to the insufficient sampling of a lens profile. Insufficient sampling causes incorrect digitalization of a lens phase function, and therefore, creating some side effects such as a focal shift and decreased focusing efficiency. For a better comparison, we further simulated the performance of a same metalens profile under different sampling conditions, using diffraction theory based on the Fraunhofer approximation (given the fact that the NA of our metalens is small). Different sampling rates (corresponding to the metalens pitch distances) were chosen to investigate their influence on the lens performance, including focal length and focusing efficiency (SI Table 2). The focusing efficiency is defined as the integrated intensity over three times of the full width at half maximum (FWHM) area in the focal plane with respect to the integrated intensity over the whole lens aperture. Our results indicate that a large pitch used for sampling the lens function can degrade the focusing performance by shifting the focal length and reducing the lens

efficiency, which becomes obvious when the pitch distance gets larger than 0.5 μm in our case.

SI Table 2. Simulated focal length and focusing efficiency of a metalens function by varying the pitch distance and pixel number of the lens with a fixed numerical aperture of 0.2 (a diameter of 100 μm and a desired focal position of 240 μm). This is equivalent to our metalens on glass designed in Fig. 2. The wavelength was fixed as 1650 nm. The used pitch distance by this paper is in bold.

Pitch distance (μm)	Focal length (μm)	Focusing efficiency
4	218	0.505
3	223	0.529
2.2	229	0.607
2	231	0.615
1	236	0.747
0.5	238	0.824
0.2	239	0.886
0.1	240	0.920

To explain the sources of loss, we have added one sentence in the main text and a Supplementary Note 7:

Page 16 in the main text:

“As such, the efficiency loss of our achromatic metalens can be attributed to two major sources: (i) a non-negligible co-polarization component in some of selected meta-atoms used for covering a broad span of group delay; (ii) insufficient sampling of a lens profile that reduces the focusing efficiency (Supplementary Note 7).”

6. Can this lens handle large laser power? Can the authors quantify the damage threshold?

Reply 6: We thank the Reviewer for the comment, which is a valid question related to the application of our metafiber in various fields of science and applications. In the following, we address this question by discussing the damage threshold in the context of femtosecond laser pulses, since our concept could be used for nonlinear frequency conversion (e.g., in dispersion-engineered fibers) of spectral broadband (i.e., femtosecond) light pulses. To address this issue in detail, we (A) estimated the damage threshold of polymers based on literature values and (B) performed additional experiments on test samples. Both investigations confirm that the use of typical ultrafast laser pulses is unproblematic in the context of our metafiber device.

A. Comparison of the fluence in the experiments performed with the polymer damage fluence:

The damage threshold of materials in the context of ultrafast laser pulses is typically defined by the damage fluence F_d (unit: J/m^2) which for polymeric material is of the order of $F_d=3500 \text{ J}/\text{m}^2$ (see for instance *AIP Conference Proceedings* **1278**, 56 (2010); <https://doi.org/10.1063/1.3507148>). The fluence F_{max} that maximally appears in the experiments presented here was estimated by dividing the maximum single-pulse energy $E_{max} = 2.13\text{nJ}$ (calculated at highest average output power $P_{ave,max}=170\text{mW}$, $\tau = 30\text{fs}$, $\nu = 80\text{MHz}$, $\lambda = 1550 \text{ nm}$) and the area of the metalens $A_m=\pi\cdot(100 \mu\text{m}/2)^2=7.85\cdot 10^{-9}\text{m}^2$, leading to $F_{max} = E_{max}/A_m= 0.27 \text{ J}/\text{m}^2$. Comparing this value with the damage fluence F_d , it can be seen that in the current experiment the damage threshold is undercut by a factor of 10000, thus clearly suggesting that polymer-based metasurfaces used here can be used in ultrafast nonlinear optics.

B. Power stability experiments on test sample: To quantify the power handling capabilities of the polymer metasurfaces, we performed additional experiments using a femtosecond fiber laser operating at $1.55\mu\text{m}$ (Toptica FemtoFiber pro IRS-II). As a test sample, we printed a planar film-type structure on a glass surface that had a diameter larger than the diameter of the metalens used in the main text. Through gently focusing the light from the ultrafast laser and placing the samples at an appropriate position after the focus (about 2mm, **Supplementary Fig. S17A**), a beam diameter of roughly $100\mu\text{m}$ was reached, matching the size of the metalens used in this work. This ensures that the intensity distributions can be quantitatively compared between these experimental runs and the metalens experiments reported in the manuscript.

Fig. S17. Laser damage threshold of a polymer metalens. (A). Experimental setup to investigate the damage threshold of nanoprinted structures. This arrangement was chosen to

achieve similar beam expansion to the fiber-based experiments. **(B)**. Recorded power at selected points of time at maximum laser power.

Via this arrangement, the nanoprinted sample was exposed to a sequence of ultrashort laser pulses over a defined duration of time and the transmitted power was recorded every minute. The measurements show that even at the maximum average output power of the laser (**Supplementary Fig. S17B**), no change in output power is observed over a period of one hour, indicating that the polymer structure is able to withstand the high intensity pulses. This observation is confirmed by additional microscopic inspection of the sample after the one-hour exposure, showing no visible change or degradation of the nanoprinted part. We would like to mention that at the maximum output power of the laser, the peak power of the single pulse is more than 50 kW, a value higher than that required in many solid-state fiber-based nonlinear frequency conversion systems (see for instance *Optics Express* **21**, 10969-10977 (2013). <https://doi.org/10.1364/OE.21.010969>).

C. Previous study on optical trapping: We would also like to mention that a different type of 3D nanoprinted metalens was used in a previous study for optical trapping experiments [Ref. 43]. In the experiment reported there, cw-laser power levels of the order of 50mW in the red spectral domain have been used, showing no degradation of the nanoprinted structure for any of the experiments (which partially went over hours), thus again confirming that metalens can withstand a substantial amount of laser power.

Based on the above points, it can be clearly stated that due to the comparatively large transverse extent of the metasurface structure (on the order of 100 μm), the local light fluence in commonly used experimental configurations is sufficiently low to prevent damage to the nanoprinted structure.

To account for the Reviewer's comment, we have slightly extended the main text and added a note to the supplementary information as follows:

Page 17 in the main text:

“Here we would like to highlight that due to the comparatively large transverse extent of the metasurface structure (on the order of 100 μm), the local light fluence in commonly used experimental configurations (e.g., nonlinear frequency conversion of optical trapping) is sufficiently low to prevent damage to the nanoprinted structure (Supplementary Note 9 and Fig. S17).”

“Supplementary Note 9. Laser damage threshold of a polymer metalens.

To assess how the nanoprinted structures are affected by high light intensities, (A) the damage threshold of polymers based on literature values was estimated and (B) additional experiments on nanoprinted test samples were performed.

A. Comparison of the fluence in the experiments performed with the polymer damage fluence: The damage threshold of materials in the context of ultrafast laser pulses is typically

defined by the damage fluence F_d (unit: J/m^2) which for polymeric material is of the order of $F_d=3500 \text{ J}/\text{m}^2$ (see for instance *AIP Conference Proceedings* **1278**, 56 (2010); <https://doi.org/10.1063/1.3507148>). The fluence F_{max} that maximally appears in the experiments presented here was estimated by dividing the maximum single-pulse energy $E_{max} = 2.13\text{ nJ}$ (calculated at highest average output power $P_{ave,max}=170\text{ mW}$, $\tau = 30\text{ fs}$, $\nu = 80\text{ MHz}$, $\lambda = 1550 \text{ nm}$) and the area of the metalens $A_m=\pi\cdot(100 \mu\text{m}/2)^2=7.85\cdot 10^{-9}\text{ m}^2$, leading to $F_{max} = E_{max}/A_m= 0.27 \text{ J}/\text{m}^2$. Comparing this value with the damage fluence F_d , it can be seen that in the current experiment the damage threshold is undercut by a factor of 10000, thus clearly suggesting that polymer-based metasurfaces used here can be used in ultrafast nonlinear optics.

B. Power stability experiments on test sample: To quantify the power handling capabilities of the polymer metasurfaces, we performed additional experiments using a femtosecond fiber laser operating at $1.55\mu\text{m}$ (Toptica FemtoFiber pro IRS-II). As a test sample, we printed a planar film-type structure on a glass surface that had a diameter larger than the diameter of the metalens used in the main text. Through gently focusing the light from the ultrafast laser and placing the samples at an appropriate position after the focus (about 2mm, Supplementary Fig. S17A), a beam diameter of roughly $100\mu\text{m}$ was reached, matching the size of the metalens used in this work. This ensures that the intensity distributions can be quantitatively compared between these experimental runs and the metalens experiments reported in the manuscript.

Via this arrangement, the nanoprinted sample was exposed to a sequence of ultrashort laser pulses over a defined duration of time and the transmitted power was recorded every minute. The measurements show that even at the maximum average output power of the laser (Supplementary Fig. S17B), no change in output power is observed over a period of one hour, indicating that the polymer structure is able to withstand the high intensity pulses. This observation is confirmed by additional microscopic inspection of the sample after the one-hour exposure, showing no visible change or degradation of the nanoprinted part. We would like to mention that at the maximum output power of the laser, the peak power of the single pulse is more than 50 kW, a value higher than that required in many solid-state fiber-based nonlinear frequency conversion systems (see for instance *Optics Express* **21**, 10969-10977 (2013). <https://doi.org/10.1364/OE.21.010969>).

C. Previous study on optical trapping: We would also like to mention that a different type of 3D nanoprinted metalens was used in a previous study for optical trapping experiments [Ref. 43]. In the experiment reported there, cw-laser power levels of the order of 50mW in the red spectral domain have been used, showing no degradation of the nanoprinted structure for any of the experiments (which partially went over hours), thus again confirming that metalens can withstand a substantial amount of laser power.

Based on the above points, it can be clearly stated that due to the comparatively large transverse extent of the metasurface structure (on the order of $100 \mu\text{m}$), the local light

fluence in commonly used experimental configurations is sufficiently low to prevent damage to the nanoprinted structure. Both investigations confirm that the use of typical ultrafast laser pulses is unproblematic in the context of our metafiber device.”

7. In Fig. 1, what are those features on the side of the 3D-nanoprinted tower?

Reply 7: On the 3D-nanoprinted hollow tower, we created some rectangular holes on the side wall, allowing the inside photoresist intact from the laser exposure to be fully removed from the chemical development process. To clarify this point, we have added one sentence in the main text:

Page 19 in the main text:

“On the 3D-nanoprinted hollow tower, we created some rectangular holes on the side wall, allowing the inside photoresist intact from the laser exposure to be fully removed from the chemical development process.”

8. P. 13 refers to Fig. 12 that doesn't exist. Please fix.

Reply 8: On Page 14 of revised main text, we have modified the description of Supplementary Fig. S14:

Page 14 in the main text:

“The optical setup used for the measurement of the fiber-optic confocal imaging is schematically illustrated in Fig. S13.”

REVIEWERS' COMMENTS

Reviewer #1 (Remarks to the Author):

The additional simulations of the metalens for off-axis imaging are very useful as well as the suggestion for a practical implementation as an endoscope. Since the focus of the paper is not endoscopy, this level of detail is sufficient.

Reviewer #2 (Remarks to the Author):

I am satisfied by the response of the authors and recommend the publication of this work in Nature communications.

Reviewer #3 (Remarks to the Author):

The authors have addressed all my comments satisfactorily. I believe the manuscript is ready for publication.